# RePaFormer: Ferocious and Scalable Acceleration of MetaFormers via Structural Reparameterization

## Abstract

We reveal that feed-forward network (FFN) layers significantly contribute to the latencies of Vision Transformers (ViTs). This effect scales up quickly as the model size escalates, and hence presents a major opportunity in efficiency optimization for ViTs via structural reparameterization on FFN layers. However, directly reparameterizing the linear projection weights is difficult due to the non-linear activation in between. In this work, we propose an innovative channel idle mechanism that establishes a linear pathway through the activation function, facilitating structural reparameterization on FFN layers during inference. Consequently, we present a family of efficient ViTs embedded with the introduced mechanism called **RePa**rameterizable Vision Trans**Formers** (RePaFormers). This technique brings remarkable latency reductions with small sacrifices (sometimes gains) in accuracy across various MetaFormer-structured architectures investigated in the experiments. The benefits of this method scale consistently with model sizes, demonstrating increasing efficiency improvements and narrowing performance gaps as model sizes grow. Specifically, the RePaFormer variants for DeiT-Base and Swin-Base achieve 67.5% and 49.7% throughput accelerations with minor changes in top-1 accuracy (-0.4% and -0.9%), respectively. Further improvements in speed and accuracy are expected on even larger ViT models. In particular, the RePaFormer variants for ViT-Large and ViT-Huge enjoy 66.8% and 68.7% inference speed-ups with +1.7% and +1.1% higher top-1 accuracies, respectively. RePaFormer is the first to employ structural reparameterization on FFN layers to expedite ViTs to our best knowledge, and we believe that it represents an auspicious direction for efficient ViTs. Codes are provided in the supplementary material.

## 1 Introduction

Vision Transformer (ViT) (Dosovitskiy et al., 2021) and its advanced variants (Touvron et al., 2021; Liu et al., 2021; Tolstikhin et al., 2021; Ryoo et al., 2021; Yu et al., 2022; Liu et al., 2022; Dehghani et al., 2023) have achieved remarkable performance on various computer vision tasks. However, the high computational cost and memory demand of ViTs hinder their wide deployment in real-world scenarios, especially in computing resource-constrained environments.

To improve efficiency for ViTs, several techniques have been proposed, such as token pruning (Rao et al., 2021; Liang et al., 2021; Kong et al., 2022a;b; Fayyaz et al., 2022) and token merging (Bolya et al., 2023; Zong et al., 2022; Marin et al., 2023; Xu et al., 2024; Kim et al., 2024) methods that gradually reduce the number of image tokens as the layer goes deep; hierarchical architectures (Fan et al., 2021; Pan et al., 2021; Liu et al., 2021; Dong et al., 2022; Ryali et al., 2023) that extract feature information at multiple scales; hybrid architectures (Mehta & Rastegari, 2022a; Chen et al., 2022a; Maaz et al., 2022; Li et al., 2022; Zhang et al., 2023) that embed efficient convolutional neural networks (CNNs) into ViTs. Meanwhile, knowledge distillation methods (Touvron et al., 2021; Hao et al., 2022; Wu et al., 2022; Chen et al., 2022b) are introduced to further optimize and improve efficient ViTs' performance. However, these efficient ViT methods overlook a powerful network simplification technique: structural reparameterization.

Structural reparameterization (Ding et al., 2019; 2021b; Zhu et al., 2023) is typically utilized in CNNs to transform a network's structure during different phases of training and testing. Specifically, structural reparameterization merges multi-branch convolutions or adjacent linear projections via

linear algebra operations (*e.g.*, aggregating parallel convolutional kernels into a single equivalent kernel). As a result, a complicated architecture during training can be compressed into a simplified structure for testing, drastically improving the model efficiency without sacrificing accuracy. Some recent research (Vasu et al., 2023; Guo et al., 2024) has explored leveraging structural reparameterization for ViTs by integrating elements from CNNs into ViTs and subsequently reparameterizing these components. However, these approaches barely reparameterize the vanilla structure of ViTs, especially the feed-forward network (FFN) layers.

Despite being less investigated, structural reparameterization holds significant potential in simplifying FFN layers for ViT and its variants. As Figure 1(a) illustrates, the FFN layer is a straightforward component, incorporating two linear transformations and an activation function in between. Although the structure is simple, the FFN layer plays an essential role in not only ViT-based models but also MetaFormer-based models (Yu et al., 2022). In these models, while the multi-head self-attention module can be replaced by other efficient token mixers like average pooling, the FFN layer remains indispensable. Moreover, some studies point out that FFN layers contribute to more than 60% total computational complexity of a ViT model (Li et al., 2022; Mehta & Rastegari, 2022b). We also observe a large portion of FFN layers in the total latency of a MetaFormer-structured model,

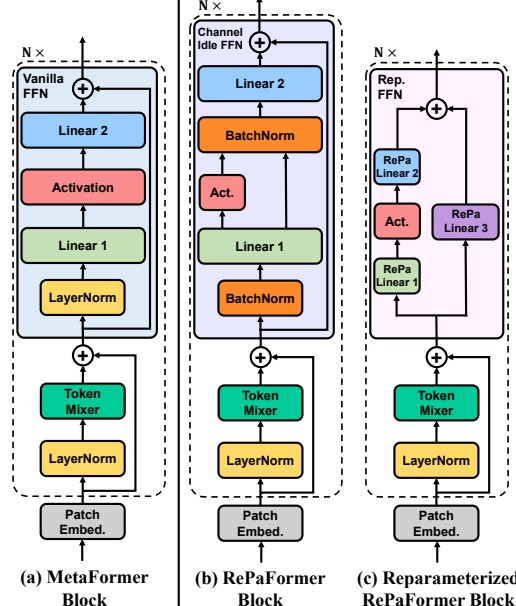

Figure 1: **RePaFormer architecture.** (a) represents the vanilla MetaFormer block, which is a general architecture for various models. For example, (a) becomes a standard ViT with self-attention as the token mixer. (b) illustrates our channel idle mechanism, where only a subset of channels are activated while the rest channels form a linear pathway. (c) shows the reparameterized RePaFormer block during testing, where the number of parameters and computational complexity are significantly reduced.

which increases as the model size grows. These factors indicate the importance of exploring approaches to enhance the efficiency of FFN layers.

However, since structural reparameterization relies on linear algebra operations to simplify the network structure, the non-linear activation function between the two linear transformations makes reparameterization infeasible on FFN layers. In addition, the LayerNorm (Lei Ba et al., 2016) in the FFN layer prevents further reparameterizing the normalization and shortcut into linear projection weights due to the sample-specific nature of LayerNorm.

To address the aforementioned challenges, we introduce an innovative channel idle mechanism. In particular, in each FFN layer, only a small subset of feature channels undergo the activation function to provide necessary nonlinearity while the rest channels remain idle, as shown in Figure 1(b). Consequently, these idle channels bridge a linear pathway through the activation function, facilitating structural reparameterization during inference. Moreover, inspired by Yao et al. (2021), we substitute the LayerNorm with BatchNorm (Ioffe & Szegedy, 2015) and add another BatchNorm before the second linear projection. These BatchNorms can be reparameterized into their corresponding linear projection weights, which further allows reparameterization of the shortcut.

With the proposed channel idle mechanism, a family of **RePa**rameterizable Vision Trans**formers** (RePaFormers) are developed, whose FFN layers can be reparameterized to condensed structures during inference as Figure 1(c) shows. RePaFormers achieve ferocious real-time accelerations of up to 133.4% post-reparameterization. Extensive experiments on various MetaFormer-structured backbones have validated the effectiveness of our method, demonstrating its potential to enhance the practical utility of MetaFormer-structured models in resource-constrained environments. Moreover, as Figure 2 illustrates, the experimental results further indicate that our method delivers more significant acceleration and narrower performance disparity as the model complexity increases, highlighting the potential of applying our method on large foundation models.

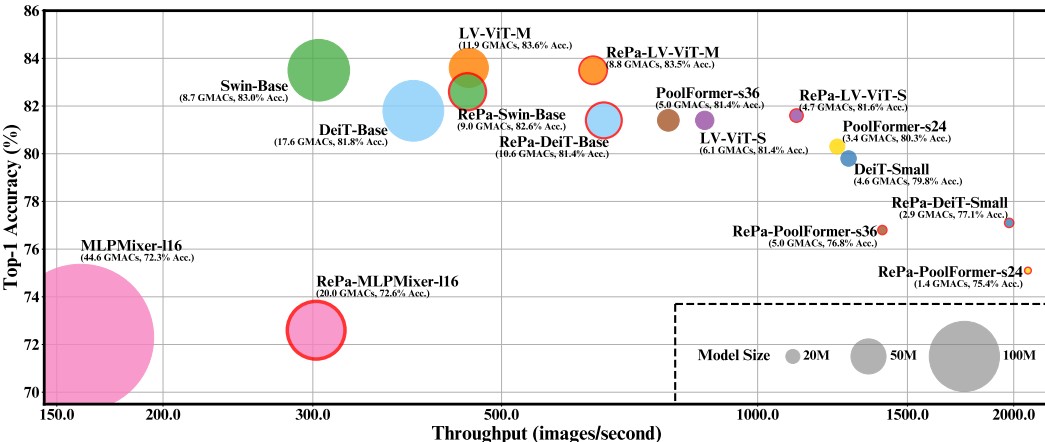

Figure 2: **Performance comparison of RePaFormers and their vanilla backbones.** RePaFormers consistently achieves more significant model accelerations and smaller accuracy gaps when the model sizes increase, highlighting its potential effectiveness in expediting large foundation models.

In conclusion, the contributions of our work are threefold: 1) We discover that FFN layers dominate in the total latency of various MetaFormer-structured models. The proportion of FFN layers in the computation significantly increases as the model size grows. 2) We propose a novel channel idle mechanism to construct a linear pathway in the FFN layer during training, which enables reparameterizing the linear projection weights during testing without accuracy loss. 3) With the proposed mechanism, we develop highly efficient RePaFormer models based on existing MetaFormer-structured architectures. Our approach achieves greater efficiency and a narrower accuracy gap when the model size escalates. To our best knowledge, RePaFormer is the first method that successfully applies structural reparameterization on FFN layers for efficient ViTs, and achieves significant acceleration (∼68%) while having positive gains in accuracy (1∼2%) instead of accuracy drops, on large and huge ViTs.

## 2 RELATED WORK

### 2.1 EFFICIENT VISION TRANSFORMER METHODS

Vision Transformer (ViT) (Dosovitskiy et al., 2021) adapts the Transformer (Vaswani et al., 2017) architecture for computer vision, achieving success on various computer vision tasks. However, ViT suffers a substantial computational complexity. To alleviate the computational burden, several techniques that focus on structural design for efficient ViTs have been proposed. Spatial-wise token reduction methods are developed to identify less important tokens and subsequently prune (Rao et al., 2021; Liang et al., 2021; Kong et al., 2022a; Fayyaz et al., 2022; Xu et al., 2022; Meng et al., 2022; Tang et al., 2022; Xu et al., 2023) or merge (Bolya et al., 2023; Zong et al., 2022; Marin et al., 2023; Xu et al., 2024; Kim et al., 2024) them during inference. As a result, the number of tokens participating in the self-attention computation is reduced. Meanwhile, hybrid architectures that combine self-attentions with computationally efficient convolutions (Graham et al., 2021; Mehta & Rastegari, 2022a; Chen et al., 2022a; Li et al., 2022; Cai et al., 2023; Vasu et al., 2023; Zhang et al., 2023; Shaker et al., 2023) are introduced to reduce the computationally expensive self-attention operations while introducing regional biases into ViTs. In addition to hybrid ViTs, MetaFormer (Yu et al., 2022) figures out that ViTs benefit from their architectural design, which consists of one token mixer layer and one multi-layer perception layer, and the token mixer can be replaced by more efficient operations, such as average pooling (Yu et al., 2022) or linear projection (Tolstikhin et al., 2021). However, these approaches overlook the structural reparameterization method, which can effectively compress a network that contains many consecutive linear transformations, such as FFN layers in ViTs. Our work is the first to apply structural reparameterization on FFN layers for ViTs.

### 2.2 STRUCTURAL REPARAMETERIZATION

Structural reparameterization is an effective network simplification technique that is typically employed in multi-branch CNNs (Ding et al., 2019; Guo et al., 2020; Ding et al., 2021a;b). It converts

an over-parameterized network block into a compressed structure during testing, thereby reducing the model complexity and increasing the speed for the inference stage. For instance, after reparameterizing its multi-branch convolutions and shortcuts into a single branch, RepVGG-B0 (Ding et al., 2021b) achieves 71% speed-up with no accuracy loss. Although some recent studies claim to adopt structural reparameterization for enhancing ViTs' efficiency (Vasu et al., 2023; Wang et al., 2024; Tan et al., 2024), they primarily construct a hybrid architecture consisting of both convolutions and self-attentions and only perform reparameterization on the convolutional part. A recent state-of-the-art method, SLAB (Guo et al., 2024), proposes to progressively substitute LayerNorms in ViTs with BatchNorms and subsequently reparameterize BatchNorms into linear projection weights. Unlike these methods, we are the first to apply structural reparameterization on FFN layers.

## 3 METAFORMER STRUCTURE AND LATENCY ANALYSIS

### 3.1 METAFORMER-STRUCTURED MODELS

We start by revisiting the MetaFormer (Yu et al., 2022) architecture, which can be regarded as a general architecture for a variety of ViT models. Specifically, given an input $X \in \mathbb{R}^{N \times C}$ to a MetaFormer block, where $N$ represents the number of tokens and $C$ denotes the number of feature channels, the input $X$ is sequentially processed by a token mixer (TokenMixer) layer and a feed-forward network (FFN) layer, with a pre-layer Layer Normalization (LN) (Lei Ba et al., 2016) and a shortcut (He et al., 2016) for each layer as

$$
\begin{aligned}
Y &= \text{TokenMixer}(\text{LN}(X)) + X, \\
Z &= \text{FFN}(\text{LN}(Y)) + Y,
\end{aligned}
\tag{1}
$$

where $Y$ is the token mixer output and $Z$ is the FFN output as well as the input to the next block. The token mixer is utilized to aggregate image tokens, which can be multi-head self-attention (Dosovitskiy et al., 2021), average pooling (Yu et al., 2022), convolution (Li et al., 2023), etc.

In each FFN layer, the LN normalized feature $Y$ is processed by two linear projections with a non-linear activation function in between as

$$
Z = \text{FFN}(\text{LN}(Y)) + Y = \text{Act}(\text{LN}(Y)W^{\text{In}})W^{\text{Out}} + Y,
\tag{2}
$$

where $W^{\text{In}} \in \mathbb{R}^{C \times \rho C}, W^{\text{Out}} \in \mathbb{R}^{\rho C \times C}$ are the linear projection weights, and $\text{Act}(\cdot)$ is usually the GELU (Hendrycks & Gimpel, 2016) activation function. $\rho$ is the FFN expansion ratio, which is typically set to 4. The biases are omitted for simplicity since they are inherently linear and do not interfere with the reparameterization process.

Figure 1(a) illustrates the MetaFormer block. We classify all the models that adhere to this specific framework as **MetaFormer-structured** models and use this term throughout this paper.

### 3.2 LATENCY ANALYSIS

To understand the significance of improving efficiency for FFN layers, we profile the latencies of major components in four representative MetaFormer-structured models and visualize the results in Figure 3. The four models include a plain-structured ViT (DeiT (Touvron et al., 2021)), a hierarchical-structured ViT (Swin Transformer (Liu et al., 2021)), a pooling-based MetaFormer (Poolformer (Yu et al., 2022)) and a spatial MLP-based MetaFormer (MLPMixer (Tolstikhin et al., 2021)). The running times of patch embedding, token mixer and FFN layers are recorded when processing a single input image. The latency profiling draws several interesting observations:

**Observation 1: The proportion of time taken by FFN layers in the total inference time escalates quickly as the model size increases.**

Figure 3 illustrates that FFN layers constitute a substantial portion of the total processing time, which rises as the model size increases. For instance, in the DeiT-Small model, FFN layers contribute to approximately 32.8% of the inference time, while in the DeiT-Base model, this proportion increases to 43.1%. This trend is consistent across various models. For example, in the MLPMixer-b16 model, FFN layers account for about 52.7% of the total time, which rises to 66.4% in the much larger MLPMixer-l16 model.

This phenomenon arises because scaling up ViTs typically involves increasing the number of channels, whereas the number of tokens tends to remain constant. Meanwhile, the computational complexity of

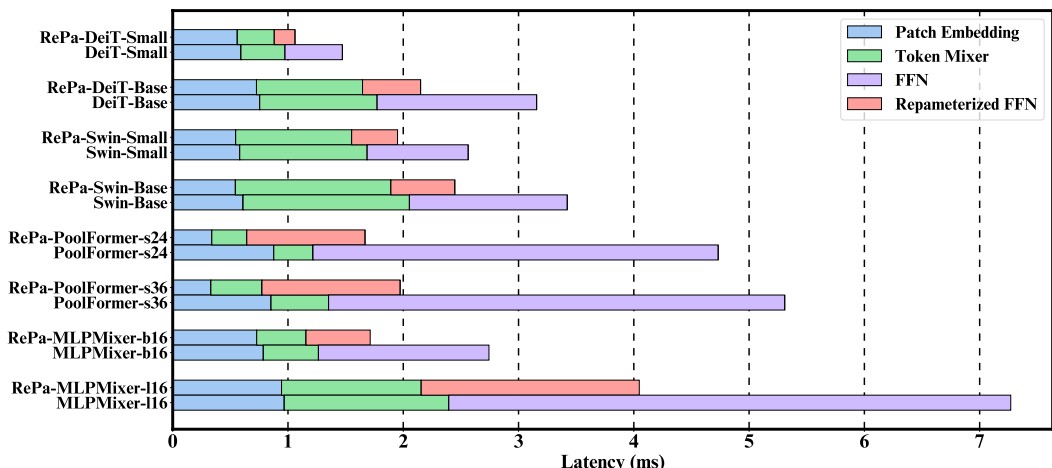

Figure 3: **Latency analysis.** Visualization of the runtime latencies of patch embedding, token mixer and FFN layers across different sizes and architectures of MetaFormer-structured models on a single GPU. Notably, as the model size increases, the proportion of latency attributed to the FFN layers also rises. Moreover, the proportion also increases with simpler token mixers. Our RePaFormer method effectively reduces the latency of FFN layers and obtains increasingly better performance on larger models, demonstrating a scalable acceleration of FFN layers.

an FFN layer, quantified as $O(2\rho NC^2)$, is quadratic to the number of feature channels. Consequently, as the model expands, the FFN layers become significantly more computationally expensive.

**Observation 2: The proportion of time taken by FFN layers in the total inference time signifies as the token mixer becomes simpler.**

Figure 3 also implies that the proportion of processing time allocated to FFN layers is affected by the simplicity of the token mixer. For instance, while the Swin-Small and Poolformer-s36 models have comparable total times for processing a single image, their FFN layers' proportions differ significantly. On the one hand, Swin-Small, which utilizes the complex self-attention mechanism as its token mixer, allocates approximately 33.6% processing time to FFN layers. On the other hand, Poolformer-s36, which employs a simpler pooling strategy for token mixing, attributes around 73.7% of its processing time to FFN layers. This contrast indicates that a simpler token mixer in a MetaFormer-structured model results in a larger portion of FFN layers in the computation.

**Remark:** Observations 1 and 2 underscore the growing demand for optimizing FFN layers as MetaFormer-structured models scale up rapidly nowadays, noting that the inference time of FFN layers predominates in the total inference time. This increasing dominance further signifies the crucial role FFN layers play in overall model performance and efficiency. Moreover, for strategies that concentrate on reducing the complexity of token mixers, enhancing the efficiency of FFN layers can lead to further acceleration. In conclusion, prioritizing the optimization of the FFN layer is of considerable importance for minimizing the overall computational costs associated with various MetaFormer-structured architectures.

## 4 METHOD

### 4.1 CHANNEL IDLE MECHANISM FOR FFN

As shown in Equation 2, due to the non-linear activation function, the structural reparameterization cannot directly merge the two linear projection weights $W^{\text{In}}$ and $W^{\text{Out}}$ via linear algebra operations.

Inspired by ShuffleNetv2 (Ma et al., 2018) which keeps a group of channels idle in grouped convolutions and shuffles channels for information exchange, we propose a simple yet effective channel idle mechanism to enable reparameterization in FFN layers. In particular, this mechanism maintains a large subset of feature channels inactivated in an FFN layer as Figure 4(a) illustrates, which subsequently bridges a linear pathway through the non-linear activation function in the corresponding FFN layer. Given that the second linear projection fuses the feature information from both activated and

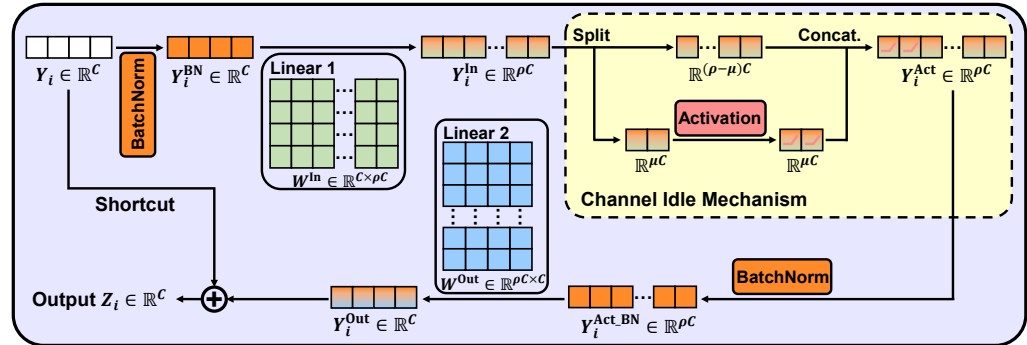

(a) **The FFN layer with channel idle mechanism (training stage).**

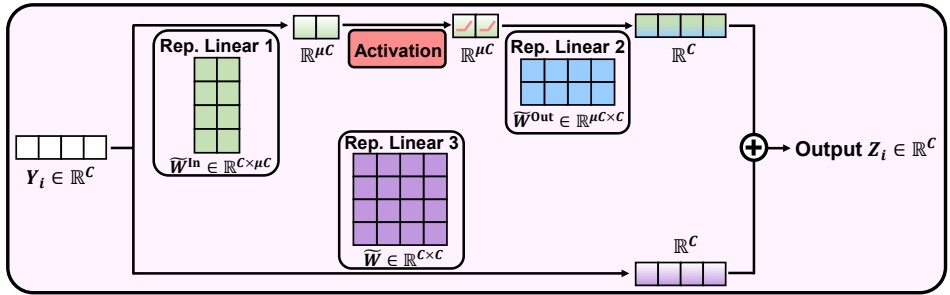

(b) **The reparameterized FFN layer (testing stage).**

Figure 4: **RePaFormer FFN architecture.** (a) illustrates the FFN layer with our proposed channel idle mechanism during training. Only a small subset of feature channels are activated while the rest keep idling. (b) shows the reparameterized FFN layer during the testing stage. Consequently, the two large linear projection weights in the training stage are reparameterized into three smaller linear projection weights, subsequently reducing the model size and computational complexity.

idling channels, there is no need to "shuffle" the channels as ShuffleNetv2 does. Our channel idle mechanism can be formulated as follows:

$$
\begin{aligned}
\boldsymbol{Y}^{\text{In}} &= \text{BN}(\boldsymbol{Y})\boldsymbol{W}^{\text{In}}, \\
\boldsymbol{Y}^{\text{Act}} &= \text{Concat}(\text{Act}(\boldsymbol{Y}^{\text{In}}_{[:,\ 1:\mu C]}), \boldsymbol{Y}^{\text{In}}_{[:,\ \mu C+1:\rho C]}), \\
\boldsymbol{Z} &= \text{BN}(\boldsymbol{Y}^{\text{Act}})\boldsymbol{W}^{\text{Out}} + \boldsymbol{Y},
\end{aligned}
\tag{3}
$$

where the activation function is only applied on $\mu C$ ($\mu < \rho$) feature channels. The $(\rho - \mu)C$ idling feature channels construct a linear route in the FFN layer. We further define the channel idle ratio as $\theta = 1 - \frac{\mu}{\rho}$, which represents the percentage of feature channels keeping idle in the activation. $\mu$ is set to 1 by default in the following experiments unless otherwise noted, leading to the default $\theta = 1 - \frac{1}{\rho}$ (*e.g.*, $\theta = 0.75$ when $\rho = 4$, indicating 75% channels are idling when the expansion ratio is 4).

### 4.2 STRUCTURAL REPARAMETERIZATION FOR FFN

With the channel idle mechanism defined in Equation 3, we are able to simplify the FFN layer by structural reparameterization during the testing stage. Firstly, we reparameterize BatchNorms into their corresponding linear projection weights as

$$
\begin{aligned}
\widetilde{\mathbf{W}}^{\text{In}} &= \frac{\gamma_{\boldsymbol{Y}}}{\sqrt{\sigma_{\boldsymbol{Y}}^2 + \epsilon_{\boldsymbol{Y}}}} \boldsymbol{W}^{\text{In}}, \\
\widetilde{\mathbf{W}}^{\text{Out}} &= \frac{\gamma_{\boldsymbol{Y}^{\text{Act}}}}{\sqrt{\sigma_{\boldsymbol{Y}^{\text{Act}}}^2 + \epsilon_{\boldsymbol{Y}^{\text{Act}}}}} \boldsymbol{W}^{\text{Out}},
\end{aligned}
\tag{4}
$$

where $\gamma$s, $\sigma^2$s and $\epsilon$s are the empirical means, empirical variances and constants from the frozen BatchNorm layers, respectively. With the reparameterized projection weights $\widetilde{\mathbf{W}}^{\text{In}}$ and $\widetilde{\mathbf{W}}^{\text{Out}}$, the

output $\boldsymbol{Z}$ in Equation 3 can now be derived by

$$\boldsymbol{Z} = \text{Act}(\boldsymbol{Y}\widetilde{\boldsymbol{W}}^{\text{In}}_{[:, \ 1:\mu C]})\widetilde{\boldsymbol{W}}^{\text{Out}}_{[1:\mu C, \ :]} + \boldsymbol{Y}\widetilde{\boldsymbol{W}}^{\text{In}}_{[:, \ \mu C+1:\rho C]}\widetilde{\boldsymbol{W}}^{\text{Out}}_{[\mu C+1:\rho C, \ :]} + \boldsymbol{Y}. \tag{5}$$

Then, we further reparameterize the weights as

$$\widetilde{\boldsymbol{W}} = \widetilde{\boldsymbol{W}}^{\text{In}}_{[:, \ \mu C+1:\rho C]}\widetilde{\boldsymbol{W}}^{\text{Out}}_{[\mu C+1:\rho C, \ :]} + I. \tag{6}$$

By substituting Equation 6 into Equation 5, we obtain the updating function for the FFN layer during the testing stage with three reparameterized weights as

$$\boldsymbol{Z} = \text{Act}(\boldsymbol{Y}\widetilde{\boldsymbol{W}}^{\text{In}}_{[:, \ 1:\mu C]})\widetilde{\boldsymbol{W}}^{\text{Out}}_{[1:\mu C, \ :]} + \boldsymbol{Y}\widetilde{\boldsymbol{W}}. \tag{7}$$

As Figure 4(b) shows, after reparameterization, the two massive linear projections are converted into three more efficient linear transformations with fewer parameters and all the normalizations are merged into linear projection weights.

## 4.3 COMPUTATIONAL COMPLEXITY ANALYSIS

**Number of parameters:** The vanilla FFN layer's parameters are mainly derived from the two linear projection weights $\boldsymbol{W}^{\text{In}} \in \mathbb{R}^{C \times \rho C}$ and $\boldsymbol{W}^{\text{Out}} \in \mathbb{R}^{\rho C \times C}$, totalling $2\rho C^2$. In contrast, with the implementation of our channel idle mechanism, the weights are reparameterized into three terms: an input weight $\widetilde{\boldsymbol{W}}^{\text{In}}_{[:, \ 1:\mu C]} \in \mathbb{R}^{C \times \mu C}$, an output weight $\widetilde{\boldsymbol{W}}^{\text{Out}}_{[1:\mu C, \ :]} \in \mathbb{R}^{\mu C \times C}$ and a reparameterized weight $\widetilde{\boldsymbol{W}} \in \mathbb{R}^{C \times C}$. The total number of parameters is effectively reduced from $2\rho C^2$ to $(2\mu + 1)C^2$.

Consequently, in the reparameterized FFN layer, the parameter count is diminished to $1 - \theta + \frac{1}{2\rho}$ of the original parameter count, where $\theta$ is the aforementioned idle ratio. For instance, when $\rho = 4$ and $\theta = 0.75$, the number of parameters in an FFN layer declines to 37.5% post-parameterization. This reduction significantly simplifies the model, diminishing its memory consumption.

**Computational complexity:** The computational complexity of the vanilla FFN layer is $O(2\rho NC^2)$ while the computational complexity is significantly reduced to $O((2\mu + 1)NC^2)$ in our reparameterized FFN layer. The computational complexity reduction ratio for an FFN layer is also $1 - \theta + \frac{1}{2\rho}$.

## 5 EXPERIMENTS

In this section, we aim to evaluate our method from the following aspects: firstly, the pre- and post-reparameterization performance of RePaFormers on the image classification task to illustrate the efficiency improvement our method brings; secondly, a competitive study of RePaFormers with their vanilla backbones to demonstrate the scalable accelerations achieved through our approach; thirdly, the comparison against a recent state-of-the-art efficient ViT method via reparameterization to demonstrate the competitive edge of our method; next, a sensitivity study on the idle ratio to emphasize the critical balance between performance and efficiency; and finally, a validation of the generalizability of RePaFormers on a self-supervised learning task and dense prediction tasks.

## 5.1 DATASETS, TRAINING AND EVALUATION SETTINGS

We mainly train and test RePaFormers for the image classification task on the widely recognized ImageNet-1k (Deng et al., 2009) dataset, following the data augmentations and training recipes proposed by Touvron et al. (2021) as the standard practice. In line with Yao et al. (2021), the maximum learning rate is set to $5 \times 10^{-3}$ with 20 epochs of warmup from $1 \times 10^{-6}$. The default batch size and total training epochs are 4096 and 300, respectively. Additionally, the Lamb optimizer (You et al., 2020) is utilized for stable training with a large batch size. For dense prediction tasks, we follow the configurations from MMDetection (Chen et al., 2019) and MMSegmentation (Contributors, 2020) to finetune RePaFormers on MSCOCO (Lin et al., 2014) and ADE20K (Zhou et al., 2017) datasets for object detection and segmentation tasks, respectively. All the models are trained from scratch on NVIDIA H100 GPUs. To ensure fair comparisons, we measure the throughput of all the models on the same NVIDIA A6000 GPU with the same environments and a fixed batch size of 128. More implementation details on the training settings are provided in the appendix.

Table 1: **Main results of RePaFormers pre- and post-reparameterization.** We employ our channel idle mechanism on various MetaFormer-structured backbones and report their accuracy, number of parameters, computational complexities and throughputs before ($\times$ for "Rep.") and after ($\sqrt{}$ for "Rep.") reparameterization. All the models become ferociously more efficient after being reparameterized **without** accuracy loss.

| Model | Rep. | Embed. dim. | Depth | #Heads | #MParam. | Complexity (GMACs) | Throughput (img/s) | Top-1 acc. |
|---|---|---|---|---|---|---|---|---|
| RePa-DeiT-Tiny | $\times$ | 192 | 12 | 3 | 5.7 | 1.1 | 2333.6 | 64.3% |
| | $\sqrt{}$ | | | | 3.5 ($-38.6\%$) | 0.6 ($-45.5\%$) | 4295.0 ($+84.1\%$) | |
| RePa-DeiT-Small | $\times$ | 384 | 12 | 6 | 22.1 | 4.3 | 1037.1 | 77.1% |
| | $\sqrt{}$ | | | | 13.2 ($-40.3\%$) | 2.5 ($-41.9\%$) | 1975.5 ($+90.5\%$) | |
| RePa-DeiT-Base | $\times$ | 768 | 12 | 12 | 86.6 | 16.9 | 336.6 | 81.4% |
| | $\sqrt{}$ | | | | 51.1 ($-41.0\%$) | 10.6 ($-37.3\%$) | 659.5 ($+95.9\%$) | |
| RePa-ViT-Large | $\times$ | 1024 | 24 | 16 | 304.5 | 59.8 | 102.7 | 82.0% |
| | $\sqrt{}$ | | | | 178.4 ($-41.4\%$) | 34.9 ($-41.6\%$) | 207.2 ($+101.8\%$) | |
| RePa-ViT-Huge | $\times$ | 1280 | 32 | 16 | 632.5 | 124.4 | 53.0 | 81.4% |
| | $\sqrt{}$ | | | | 369.9 ($-41.5\%$) | 72.6 ($-41.6\%$) | 103.8 ($+95.8\%$) | |
| RePa-Swin-Tiny | $\times$ | [96, 192, 384, 768] | [2, 2, 6, 2] | [3, 6, 12, 24] | 28.3 | 4.4 | 611.8 | 78.5% |
| | $\sqrt{}$ | | | | 17.5 ($-38.2\%$) | 2.6 ($-40.9\%$) | 1026.5 ($+59.6\%$) | |
| RePa-Swin-Small | $\times$ | [96, 192, 384, 768] | [2, 2, 18, 2] | [3, 6, 12, 24] | 49.7 | 8.6 | 367.6 | 81.6% |
| | $\sqrt{}$ | | | | 29.9 ($-39.8\%$) | 5.1 ($-40.7\%$) | 624.0 ($+69.7\%$) | |
| RePa-Swin-Base | $\times$ | [128, 256, 512, 1024] | [2, 2, 18, 2] | [4, 8, 16, 32] | 87.9 | 15.2 | 250.1 | 82.6% |
| | $\sqrt{}$ | | | | 52.8 ($-39.9\%$) | 9.0 ($-40.8\%$) | 456.0 ($+82.3\%$) | |
| RePa-LV-ViT-S | $\times$ | 384 | 16 | 6 | 26.2 | 6.1 | 725.4 | 81.6% |
| | $\sqrt{}$ | | | | 19.1 ($-27.1\%$) | 4.7 ($-23.0\%$) | 1110.9 ($+53.1\%$) | |
| RePa-LV-ViT-M | $\times$ | 512 | 20 | 8 | 55.9 | 11.9 | 396.6 | 83.6% |
| | $\sqrt{}$ | | | | 40.1 ($-28.3\%$) | 8.8 ($-26.1\%$) | 640.6 ($+61.5\%$) | |
| RePa-PoolFormer-s12 | $\times$ | [64, 128, 320, 512] | [2, 2, 6, 2] | n/a | 11.9 | 1.8 | 1882.2 | 70.5% |
| | $\sqrt{}$ | | | | 6.0 ($-49.6\%$) | 0.8 ($-55.6\%$) | 3973.9 ($+111.1\%$) | |
| RePa-PoolFormer-s24 | $\times$ | [64, 128, 320, 512] | [4, 4, 12, 4] | n/a | 21.4 | 3.4 | 957.6 | 75.4% |
| | $\sqrt{}$ | | | | 9.6 ($-55.1\%$) | 1.4 ($-58.8\%$) | 2078.4 ($+117.0\%$) | |
| RePa-PoolFormer-s36 | $\times$ | [64, 128, 320, 512] | [6, 6, 18, 6] | n/a | 30.9 | 5.1 | 642.1 | 76.8% |
| | $\sqrt{}$ | | | | 13.1 ($-57.6\%$) | 2.0 ($-60.8\%$) | 1401.6 ($+119.1\%$) | |
| RePa-MLPMixer-b16 | $\times$ | 768 | 12 | n/a | 59.9 | 12.7 | 420.9 | 72.1% |
| | $\sqrt{}$ | | | | 24.4 ($-59.2\%$) | 5.7 ($-55.1\%$) | 968.4 ($+130.1\%$) | |
| RePa-MLPMixer-l16 | $\times$ | 1024 | 24 | n/a | 208.4 | 44.7 | 129.7 | 72.6% |
| | $\sqrt{}$ | | | | 82.2 ($-60.6\%$) | 20.0 ($-55.3\%$) | 302.7 ($+133.4\%$) | |

## 5.2 CLASSIFICATION RESULTS

We choose five different MetaFormer-structured backbones, including a plain-structured ViT (ViT (Dosovitskiy et al., 2021) and DeiT (Touvron et al., 2021)), a hierarchical-structured ViT (Swin Transformer (Liu et al., 2021)), a plain ViT trained with token labelling (LV-ViT (Jiang et al., 2021)), a pooling-based MetaFormer (PoolFormer (Yu et al., 2022)) and a spatial MLP-based MetaFormer (MLPMixer (Tolstikhin et al., 2021)). The FFN layers in these models are embedded with the channel idle mechanism and are all trained from scratch solely on the ImageNet-1k dataset.

### 5.2.1 REPARAMETERIZATION RESULTS

Table 1 presents the image classification performance of RePaFormers before and after reparameterization. Our innovative channel idle mechanism remarkably enhances these models' computational efficiency and throughput while preserving their accuracy. Specifically, when employing the standard ViT as the backbone, RePa-ViT-Large achieves a substantial speed-up of 101.8% with an accuracy of 82.0% post-reparameterization during the testing phase. In the hierarchical ViT architecture, RePa-Swin-Base achieves an 82.3% increase in speed after reparameterization, with an accuracy of 82.6%. For models utilizing simpler token mixers, RePaPoolformer-s36 and RePa-MLPMixer-l16 realize remarkable accelerations of 119.1% and 133.4%, respectively.

It is worth highlighting that **as the model size increases, our method yields more substantial accelerations and more significant reductions in parameters after reparameterization**. Such efficiency improvement is also illustrated in Figure 3, where the same backbone architecture obtains more speed-up when its model size escalates. This characteristic is increasingly vital as the trend towards larger foundation models in the current research community continues to grow.

### 5.2.2 COMPARISON WITH VANILLA BACKBONES

Table 2 compares the performance of RePaFormers and their original backbones. We report the accuracies that are directly trained from scratch on the ImageNet-1k training set. Overall, our method demonstrates greater acceleration on these backbones. Moreover, we point out that with the same

Table 2: **Performance comparisons among RePaFormers and their vanilla backbones.** When the token mixer architecture fixes, our method consistently achieves more accelerations and complexity reductions while narrowing the accuracy gap as the model size grows.

| Model | #MParam. | Complexity (GMACs) | Throughput (img/s) | Top-1 acc. |
|---|---|---|---|---|
| DeiT-Tiny | 5.7 | 1.3 | 3239.4 | 72.1% |
| RePa-DeiT-Tiny | 3.5 (−38.6%) | 0.8 (−38.5%) | 4295.0 (+32.6%) | 64.2% |
| DeiT-Small | 22.1 | 4.6 | 1279.1 | 79.8% |
| RePa-DeiT-Small | 13.2 (−40.3%) | 2.9 (−37.0%) | 1975.5 (+54.4%) | 77.1% |
| DeiT-Base | 86.6 | 17.6 | 393.8 | 81.8% |
| RePa-DeiT-Base | 51.1 (−41.0%) | 10.6 (−39.8%) | 659.5 (+67.5%) | 81.4% |
| ViT-Large | 304.3 | 59.7 | 124.2 | 80.3% |
| RePa-ViT-Large | 178.4 (−41.4%) | 34.9 (−41.5%) | 207.2 (+66.8%) | 82.0% |
| ViT-Huge | 632.2 | 124.3 | 61.5 | 80.3% |
| RePa-ViT-Huge | 369.9 (−41.5%) | 72.6 (−41.6%) | 103.8 (+68.7%) | 81.4% |
| Swin-Tiny | 28.3 | 4.5 | 751.9 | 81.2% |
| RePa-Swin-Tiny | 17.5 (−38.2%) | 2.6 (−42.2%) | 1026.5 (+36.5%) | 78.5% |
| Swin-Small | 49.6 | 8.7 | 441.8 | 83.0% |
| RePa-Swin-Small | 29.9 (−39.7%) | 5.1 (−41.4%) | 624.0 (+41.2%) | 81.6% |
| Swin-Base | 87.8 | 15.2 | 304.9 | 83.5% |
| RePa-Swin-Base | 52.8 (−39.9%) | 9.0 (−40.8%) | 456.0 (+49.6%) | 82.6% |
| LV-ViT-S | 26.2 | 6.1 | 866.6 | 81.4% |
| RePa-LV-ViT-S | 19.1 (−27.1%) | 4.7 (−23.0%) | 1110.9 (+28.2%) | 81.6% |
| LV-ViT-M | 55.8 | 11.9 | 457.6 | 83.6% |
| RePa-LV-ViT-M | 40.1 (−28.1%) | 8.8 (−26.1%) | 640.6 (+40.0%) | 83.5% |
| PoolFormer-s12 | 12.0 | 1.9 | 2531.5 | 77.2% |
| RePa-PoolFormer-s12 | 6.0 (−50.0%) | 0.8 (−57.9%) | 3973.9 (+57.0%) | 70.5% |
| PoolFormer-s24 | 21.4 | 3.4 | 1240.6 | 80.3% |
| RePa-PoolFormer-s24 | 9.6 (−55.1%) | 1.4 (−58.8%) | 2078.4 (+67.5%) | 75.4% |
| PoolFormer-s36 | 30.9 | 5.0 | 785.3 | 81.4% |
| RePa-PoolFormer-s36 | 13.1 (−57.6%) | 2.0 (−60.0%) | 1401.6 (+78.5%) | 76.8% |
| MLPMixer-b16 | 59.9 | 12.6 | 554.1 | 76.6% |
| RePa-MLPMixer-b16 | 24.4 (−59.2%) | 5.7 (−54.8%) | 968.4 (+74.8%) | 72.1% |
| MLPMixer-l16 | 208.2 | 44.6 | 160.0 | 72.3% |
| RePa-MLPMixer-l16 | 82.2 (−60.5%) | 20.0 (−55.2%) | 302.7 (+89.2%) | 72.6% |

Table 3: **Comparison against the state-of-the-art reparameterization method for ViTs.** With a similar number of parameters, RePaFormers obtains both faster inference speeds and higher accuracies than SLAB.

| Model | #MParam. | Compl. (GMACs) | Speed (img/s) | Top-1 acc. |
|---|---|---|---|---|
| SLAB-DeiT-Base | 86.6 | 17.1 | 387.0 | 78.9% |
| RePa-DeiT-Base (25%) | **79.5** | **15.5** | **452.2** | **81.1%** |
| SLAB-Swin-Base | 87.7 | 15.4 | 299.9 | 83.6% |
| RePa-Swin-Base (25%) | **80.8** | **14.0** | **356.3** | **83.7%** |

Table 4: **Sensitivity of channel idle ratio.** We report the performance of two RePaFormers with different channel idle ratios ($\theta$). *vanilla represents the vanilla backbone with no channel idling in FFN layers. The results show a significant accuracy drop when $\theta$ surpasses 75%.

| Model | $\theta$ | #MParam. | Compl. (GMACs) | Speed (img/s) | Top-1 acc. |
|---|---|---|---|---|---|
| RePa-DeiT-Base | 100% | 37.0 | 7.1 | 858.2 | 73.7% |
| | 75% | 51.1 | 10.6 | 657.0 | 81.4% |
| | 50% | 65.3 | 12.7 | 544.4 | 81.4% |
| | 25% | 79.5 | 15.5 | 452.2 | 81.1% |
| | *vanilla | 86.6 | 17.6 | 408.8 | 81.8% |
| RePa-Swin-Base | 100% | 38.8 | 6.5 | 539.0 | 75.5% |
| | 75% | 52.8 | 9.0 | 467.6 | 82.6% |
| | 50% | 66.8 | 11.5 | 404.5 | 83.4% |
| | 25% | 80.8 | 14.0 | 356.3 | 83.7% |
| | *vanilla | 87.8 | 15.2 | 324.9 | 83.5% |

backbone architecture, the accuracy gap between a RePaFormer and its vanilla backbone significantly narrows as the model size increases. For example, employing DeiT as the backbone, the smaller DeiT-Tiny model witnesses a 32.6% speed-up at the cost of a 7.9% accuracy loss. However, when scaled up to the DeiT-Base model, our approach delivers a 67.5% throughput improvement, with only a marginal 0.4% drop in accuracy. This pattern is consistent across various models. In cases where the backbones include additional regularizations during training, our method not only accelerates performance but also preserves accuracy to a remarkable extent. In particular, on the LV-ViT model, we facilitate a 40.0% increase in the inference speed with a negligible 0.1% decrease in accuracy.

It is also worth emphasizing that **our method yields ~68% speed-up and even 1~2% higher accuracy on large and huge ViT models**, indicating its potential on large-scale foundation models.

### 5.2.3 PERFORMANCE COMPARISON AGAINST STATE-OF-THE-ART

Table 3 compares our RePaFormer approach against SLAB (Guo et al., 2024), a recent state-of-the-art method introducing progressive reparameterized BatchNorms for ViTs. For fair comparisons with similar model sizes, the performance of RePaFormers with a 25% idle ratio is used. The results indicate that our reparameterization strategy offers a better trade-off between efficiency and accuracy. For example, when utilizing DeiT-Base as the backbone, our method not only achieves a higher speed and fewer parameters but also surpasses SLAB by a 2.2% higher accuracy.

### 5.3 SENSITIVITY OF CHANNEL IDLE RATIO

In Section 4.1, we have defined the channel idle ratio $\theta$ as the percentage of feature channels keeping idle in the activation. Table 4 illustrates the influence of $\theta$ on the performance of RePaFormers. In general, a larger $\theta$ represents more channels idling in the FFN layer, leading to a smaller number of parameters, a lower computational complexity, and a higher inference speed post-reparameterization.

Remarkably, when $\theta$ exceeds 75% which is the default idle ratio for RePaFormers, there is an obvious decline in the top-1 accuracy of both RePa-DeiT-Base and RePa-Swin-Base. For instance, when setting $\theta$ to 100% (*i.e.*, no channels being activated), the RePa-DeiT-Base's accuracy drops from 81.8% to 73.7%. Similarly, the RePa-Swin-Base model witnesses its accuracy decline from 83.5% to

75.5% with $\theta = 100\%$. This outcome demonstrates that while reducing the proportion of non-linear components can significantly enhance the model's efficiency, preserving sufficient non-linearities is also crucial for maintaining performance. It highlights the delicate balance between optimizing for speed and ensuring the robustness and accuracy of the model. In addition, we provide the results of different channel idle ratios on tiny-size models in Appendix C.

## 5.4 SELF-SUPERVISED LEARNING RESULTS

To demonstrate the generalizability of our method, we apply the channel idle mechanism to ViTs trained with self-supervised learning methods and report their performance in Table 5. Specifically, two RePaFormers based on ViT-Small and ViT-Base (Dosovitskiy et al., 2021) are trained following the self-supervised learning strategies outlined in DINO (Caron et al., 2021). Even with self-supervised learning, RePaFormers still exhibit substantial efficiency enhancement.

Notably, there is a consistent trend as observed in Section 5.2.2 that when the model size increases, our method yields greater speed improvements and a smaller accuracy gap. For example, RePa-ViT-Small achieves a 39.4% increase in speed with a 2.6% drop in accuracy when using a linear classifier. In the case of employing a larger backbone model, RePa-ViT-

Table 5: **Self-supervised learning results.**

| Model | #MParam. | Compl. (GMACs) | Speed (img/s) | $k$-NN top-1 acc. | Linear top-1 acc. |
|---|---|---|---|---|---|
| ViT-Small | 21.7 | 4.3 | 1277.0 | 72.8% | 77.0% |
| RePa-ViT-Small | 12.8 | 2.5 | 1779.6 | 69.6% | 74.4% |
| ViT-Base | 85.8 | 16.9 | 396.2 | 76.1% | 78.2% |
| RePa-ViT-Base | 50.4 | 9.9 | 623.0 | 74.1% | 77.0% |

Base realizes a more significant acceleration of 57.2% with a smaller accuracy loss of 1.2%. These results indicate a high adaptability of our RePaFormer using different learning paradigms.

## 5.5 DENSE PREDICTIONS

Table 6 presents the results of two downstream tasks. Firstly, the ImageNet-1k pre-trained RePa-Swin models are integrated with a one-stage detector RetinaNet (Lin et al., 2017) and a two-stage detector Mask R-CNN (He et al., 2017) for the object detection task on the MSCOCO dataset with $1\times$ training schedule (*i.e.*, 12 epochs). Remarkably, our RePa-Swin-Base model achieves up to 18.7% latency reduction at even a higher average precision (AP) with RetinaNet when compared to its vanilla backbone. RePA-Swin-Base also obtains a similar performance with 16.0% less latency with Mask R-CNN. Secondly, UperNet (Xiao et al., 2018) is leveraged for the semantic segmentation task on the ADE20K dataset with RePa-Swin models as backbones. Similarly, RePa-Swin-Base achieves 15.4% latency reduction with merely 1.2% mIoU loss.

Overall, the experimental results on downstream tasks reflect a consistent trend that the performance disparities are narrowing and the acceleration gains are escalating as the backbone model sizes grow. This trend on dense prediction tasks aligns with the observations in Section 5.2.2 well, which further proves the scalable acceleration capability of our channel idle mechanism.

Table 6: **Performance on dense prediction tasks.** Results on the $1\times$ training schedule are presented. The latencies (ms) per image are reported for throughput comparisons.

| Backbone | RetinaNet | | | | | | | Mask R-CNN | | | | | | | UperNet | |
|---|---|---|---|---|---|---|---|---|---|---|---|---|---|---|---|---|
| | Latency (ms) | AP | $AP_{50}$ | $AP_{75}$ | $AP_S$ | $AP_M$ | $AP_L$ | Latency (ms) | AP | $AP_{50}$ | $AP_{75}$ | $AP_S$ | $AP_M$ | $AP_L$ | Latency (ms) | mIoU |
| Swin-Small | 61.7 | 37.2 | 56.9 | 39.6 | 22.4 | 40.5 | 49.4 | 62.5 | 45.5 | 67.8 | 49.9 | 28.6 | 49.2 | 60.4 | 36.3 | 47.6 |
| RePa-Swin-Small | 53.8 $_{(-12.8\%)}$ | 38.3 | 57.9 | 40.7 | 21.8 | 42.0 | 51.6 | 53.8 $_{(-13.9\%)}$ | 43.6 | 65.8 | 47.8 | 27.1 | 47.0 | 57.3 | 32.1 $_{(-11.6\%)}$ | 45.7 |
| Swin-Base | 82.0 | 38.9 | 59.5 | 41.3 | 24.3 | 43.6 | 54.4 | 82.6 | 45.8 | 67.6 | 50.3 | 28.7 | 48.9 | 61.7 | 45.6 | 48.1 |
| RePa-Swin-Base | 66.7 $_{(-18.7\%)}$ | 39.8 | 60.0 | 42.1 | 25.3 | 43.7 | 53.8 | 69.4 $_{(-16.0\%)}$ | 44.8 | 67.0 | 49.4 | 29.0 | 48.5 | 58.4 | 38.6 $_{(-15.4\%)}$ | 46.9 |

## 6 CONCLUSION

In this paper, we investigate the latency compositions of various MetaFormer-structured models and observe that FFN layers significantly contribute to the overall latency. The observations highlight the critical need for accelerating FFN layers to enhance the efficiency of ViTs, where structural reparameterization emerges as a potential solution. We introduce a novel channel idle mechanism to facilitate the reparameterization of FFN layers during inference. The proposed mechanism is employed on various MetaFormer-structured models, resulting in a family of RePaFormers. RePaFormers demonstrate consistent scalability with more accelerations and narrower accuracy disparities as the backbone model size escalates. Importantly, RePaFormer achieves accuracy gains while improving the inference speed on large-scale ViT backbone. We believe that RePaFormer presents a promising direction for expediting ViTs and we invite the community to further explore its effectiveness on even larger MetaFormer-structured foundation models.

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

## A    TRAINING SETTINGS

All RePaFormers are rigorously trained on the ImageNet-1k dataset (Deng et al., 2009), following the same data augmentations proposed by DeiT (Touvron et al., 2021). Consistently, the total number of training epochs is standardized at 300. In an effort to accommodate the substitution of LayerNorm with BatchNorm, we have increased the batch size to 4096. Additionally, the Lamb optimizer You et al. (2020) has been selected to ensure stable training with a large batch size. Learning rates are dedicatedly tuned for different backbone architectures, and a cosine scheduler Loshchilov & Hutter (2017) is utilized for learning rate adjustment throughout the training period. Detailed training settings are provided in Table 7.

Table 7: Training settings of RePaFormers for the image classification task.

| Model | Epochs | Batch size | Optimizer | Base learning rate | Min learning rate | Warmup learning rate | Scheduler | Weight decay | Drop path rate |
|---|---|---|---|---|---|---|---|---|---|
| RePa-DeiT-Tiny | | | | $2.5 \times 10^{-3}$ | $1 \times 10^{-6}$ | | | 0.02 | 0.02 |
| RePa-DeiT-Small | | | | $3 \times 10^{-3}$ | $4 \times 10^{-5}$ | | | 0.05 | 0.04 |
| RePa-DeiT-Base | | | | $4 \times 10^{-3}$ | $4 \times 10^{-5}$ | | | 0.07 | 0.10 |
| RePa-Swin-Tiny | | | | $5 \times 10^{-3}$ | $5 \times 10^{-5}$ | | | 0.20 | 0.10 |
| RePa-Swin-Small | | | | $6 \times 10^{-3}$ | $5 \times 10^{-5}$ | | | 0.15 | 0.09 |
| RePa-Swin-Base | | 4096 | | $4 \times 10^{-3}$ | $2 \times 10^{-5}$ | | | 0.10 | 0.08 |
| RePa-PoolFormer-s12 | 300 | | Lamb | $2.5 \times 10^{-3}$ | $6 \times 10^{-6}$ | $1 \times 10^{-6}$ | Cosine scheduler | 0.16 | 0.02 |
| RePa-PoolFormer-s24 | | | | $3.5 \times 10^{-3}$ | $1.5 \times 10^{-6}$ | | | 0.08 | 0.01 |
| RePa-PoolFormer-s36 | | | | $5.5 \times 10^{-3}$ | $3 \times 10^{-6}$ | | | 0.01 | 0.03 |
| RePa-MLPMixer-b16 | | | | $5 \times 10^{-3}$ | $5 \times 10^{-6}$ | | | 0.13 | 0.04 |
| RePa-MLPMixer-l16 | | | | $4 \times 10^{-3}$ | $1 \times 10^{-5}$ | | | 0.16 | 0.05 |
| RePa-LV-ViT-S | | 1024 | | $1 \times 10^{-3}$ | $1 \times 10^{-5}$ | | | 0.05 | 0.10 |
| RePa-LV-ViT-M | | | | $1 \times 10^{-3}$ | $1 \times 10^{-5}$ | | | 0.05 | 0.10 |

Table 8: **Sensitivity of channel idle ratio.** We report the performance of two RePaFormers with different channel idle ratios ($\theta$). *vanilla represents the vanilla backbone with no channel idling in FFN layers. The results show a significant accuracy drop when $\theta$ surpasses 75%.

| Model | $\theta$ | #MParam. | Compl. (GMACs) | Speed (img/s) | Top-1 acc. |
|---|---|---|---|---|---|
| RePa-DeiT-Tiny | 75% | 3.5 | 0.8 | 4323.8 | 64.2% |
| | 50% | 4.4 | 1.0 | 3904.2 | 69.2% |
| | 25% | 5.3 | 1.2 | 3555.1 | 71.9% |
| | *vanilla | 5.7 | 1.3 | 3372.2 | 72.1% |
| RePa-Swin-Tiny | 75% | 17.5 | 2.6 | 1016.3 | 78.5% |
| | 50% | 21.8 | 3.3 | 927.8 | 80.5% |
| | 25% | 26.1 | 4.0 | 864.9 | 81.4% |
| | *vanilla | 28.3 | 4.5 | 789.8 | 81.2% |
| RePa-PoolFormer-s12 | 75% | 6.0 | 0.8 | 4000.2 | 70.5% |
| | 50% | 8.4 | 1.2 | 3345.4 | 74.3% |
| | 25% | 10.7 | 1.6 | 2910.1 | 76.8% |
| | *vanilla | 12.0 | 1.9 | 2450.0 | 77.2% |

## B    LIMITATIONS

Despite the exceptional performance of RePaFormers on large backbone models, there is a notable decrease in accuracy as the model size shrinks. For example, as demonstrated in Table 2, the accuracy of RePa-DeiT-Tiny decreases significantly from 72.1% to 64.2%. This performance drop is primarily attributed to the reduced nonlinearity in the backbone, which is a consequence of keeping channels idle. In smaller models, both the number of layers and the number of feature channels are limited, resulting in substantially fewer activated channels compared to larger models. After applying the channel idle mechanism with a high idle ratio (*e.g.*, 75%), tiny models would lack sufficient non-linear transformations. However, as the model size increases, both the number of layers and feature channels expand, enhancing the model's robustness and mitigating the impact of reduced nonlinearity.

In conclusion, while our method may not be optimally suited for tiny models, it significantly enhances the performance of large MetaFormer-structured models. We sincerely invite the research community to further investigate and validate the effectiveness of our approach on large foundational models, such as SAM (Kirillov et al., 2023) or GPT (Radford et al., 2019; Brown et al., 2020). This exploration could provide valuable insights into the scalability and adaptability of our method across various advanced computational frameworks.

## C    SENSITIVITY OF CHANNEL IDLE RATIO ON TINY MODELS

As explained in Appendix B, tiny-size models are less robust and rely on sufficient nonlinearities for a decent feature extraction capability. To validate this, we further present the performance of tiny-size ViT models with various idle ratios. As Table 8 shows, our RePaFormers demonstrate narrow performance gaps on smaller models when the idle ratio is less rigorous (i.e., $\theta = 25\%$). While scaling to small or tiny-sized models is not the primary focus of this work, our method still shows effectiveness in these cases.

## D    COMPARISON AGAINST REPVGG-STYLE REPARAMETERIZATION

The differences between our structural reparameterization method and RepVGG-style (Ding et al., 2021b) structural reparameterization are threefold:

1. **Different reparameterization solutions:** The key difference is that RepVGG reparameterizes **horizontally** across parallel convolutional kernels, while RePaFormer reparameterizes **vertically** on consecutive linear projection weights.

For instance, RepVGG reparameterizes two **parallel** convolutional branches with kernels $W_1^{\mathbf{Conv}}$ and $W_2^{\mathbf{Conv}}$ by summing them:

$$W_{\mathbf{Rep}}^{\mathbf{Conv}} = W_1^{\mathbf{Conv}} + W_2^{\mathbf{Conv}}. \tag{8}$$

On the contrary, as demonstrated in Equation 6, RePaFormer reparameterizes two **consecutive** projection weights $W_1^{\mathbf{FFN}}$ and $W_2^{\mathbf{FFN}}$ by multiplying them:

$$W_{\mathbf{Rep}}^{\mathbf{FFN}} = W_1^{\mathbf{FFN}} \cdot W_2^{\mathbf{FFN}}. \tag{9}$$

(In the above example, we omit the BatchNorm and suppose $W_1^{\mathbf{Conv}}$ and $W_2^{\mathbf{Conv}}$ have been padded to the same shape.)

2. **Different target components:** RepVGG and RepVGG-style methods apply reparameterization to multi-branch convolutional layers in CNNs, while our RePaFormer targets FFN layers in ViTs. Their application targets are distinct.

3. **Different scopes:** Although some previous works (Vasu et al., 2023; Wang et al., 2024) have attempted to adapt RepVGG-style reparameterization on ViTs by incorporating multi-branch convolutions into the ViT backbone, they only reparameterize the convolutional parts. The main scope of these works is to construct novel mobile-friendly architectures. In contrast, our method is the first to apply structural reparameterization to FFN layers and accelerate existing ViTs/MetaFormers of all sizes.

Moreover, our channel idle mechanism cannot be regarded as a special case of a dual-branch structure in RepVGG. In RepVGG, all branches must be linear so that they can be reparameterized, whereas in our approach, one branch is linear while the other one is nonlinear.

