# OpenReview forum: "RePaFormer: Ferocious and Scalable Acceleration of MetaFormers via Structural Reparamterization"
_ICLR.cc/2025/Conference — Submitted to ICLR 2025_

### Official Review · Reviewer_ufAp · 2024-11-02

**Soundness:** 2
**Presentation:** 3
**Contribution:** 2
**Rating:** 5
**Confidence:** 4

**Summary:**

The paper proposes RePaFormer, a novel approach that leverages a channel idle mechanism to enable structural reparameterization of Feed-Forward Network (FFN) layers during inference. Experiments on Vision Transformer families show its improved inference speed compared to baselines with minor performance loss.

**Strengths:**

1. The paper is well-written and well-motivated.

**Weaknesses:**

The major concern of the paper is that the current experimental setup raises concerns about the effectiveness of the proposed method.
1. **The effects of BatchNorm.**  Specifically, if I understand correctly, the vanilla backbone uses LayerNorm while RePaFormer family uses BatchNorm. It is unclear whether BatchNorm alone could improve the test performance, i.e., accuracy, of the vanilla backbone.
2. **Similarly, the effectiveness of channel idle mechanism is inadequately tested.** Specifically, consider the default case where $\mu=1$, and 75% percent of the features are idle. This implies for the RePa Linear 3 (Figure 1), the features go through a linear transformation $W_2W_1$ where $W_1 \in \mathbb{R}^{3C \times C}$ and $W_2 \in \mathbb{R}^{C \times 3C}$. However, such transformation can be represented by a $C \times C$ matrix, suggesting that the models use $6C^2$ parameters to learn a linear function that can be represented by just $C^2$ parameters. That means, RePaFormer will be useful only if it is much better in terms of accuracy than the baseline where the channel idle part is processed by a single linear layer with weight  $W \in \mathbb{R}^{C \times C}$. More specifically, this baseline should be in the form of Figure (b) [without BatchNorm inference reparameterization] and test its throughput and performance.

**Questions:**

See my weaknesses part.

---

> ### Author Response · Authors · 2024-11-15
> **Response to Reviewer ufAp**
>
> We thank Reviewer ufAp for the valuable comments concerning the two major designs of our RePaFormer models. We would like to provide clarification on these points below:
>
> ---
>
> __1. Response to W1__ ___(whether BatchNorm alone could improve the test performance)___:
>
> This is a good question. In fact, a previous work [1] has already investigated leveraging BatchNorm instead of LayerNorm in ViT and discovered that directly applying BatchNorm in ViT would lead to irregular training crashes. To address this, [1] proposes a novel approach (FFNBN) to enable training ViT with BatchNorm. However, all the experiment results on both DeiT and Swin Transformer demonstrate that __BatchNorm consistently yields worse performance than LayerNorm on ViTs__. We cite the results from [1] in the table below:
>
> |Model|Normalization|Top-1 acc|
> |:-|-|-|
> |DeiT-S|LayerNorm|__79.8%__|
> |DeiT-S|BatchNorm+FFNBN|78.8%|
> |Swin-T|LayerNorm|__81.2%__|
> |Swin-T|BatchNorm+FFNBN|80.9%|
> |Swin-S|LayerNorm|__83.0%__|
> |Swin-S|BatchNorm+FFNBN|82.8%|
> |Swin-B|LayerNorm|__83.3%__|
> |Swin-B|BatchNorm+FFNBN|83.1%|
>
> ---
>
> __2. Response to W2__ ___(whether channel idle mechanism can be replaced by a single learnable linear function)___:
>
> We thank the Reviewer for this insightful comment and believe this comment aligns with Reviewer nzL3's W2. We would like to interpret this weakness as comparing the RePaFormers:
>
>   1. trained with $6C^2$ weights and then reparameterized into $C^2$
>   2. trained with a single $C^2$ linear projection weight from scratch
>
> The table below shows the results of RePa-DeiT, RePa-Swin and RePa-LV-ViT when trained with a single $C^2$ linear projection weight. The training settings are the same as their counterparts outlined in the manuscript. After reparameterizing the BatchNorm, the differences in the inference speed and the number of parameters for these two cases should be negligible, so we only report their accuracies. The _NaN_ values represent that the training crashes.
>
> |Backbone|$6C^2$ weights + channel idle|$C^2$ linear projection weight|
> |:-|-|-|
> |RePa-DeiT-Tiny|__64.3__|59.6|
> |RePa-DeiT-Small|__77.1__|75.0|
> |RePa-DeiT-Base|__81.4__|_NaN_|
> |RePa-Swin-Tiny|__78.5__|77.1|
> |RePa-Swin-Small|__81.6__|79.3|
> |RePa-Swin-Base|__82.6__|79.6|
> |RePa-LV-ViT-S|__81.6__|_NaN_|
> |RePa-LV-ViT-M|__83.6__|_NaN_|
>
> As shown in the table, training with $6C^2$ parameters and subsequently reparameterizing them into $C^2$ __consistently achieves better performance__, in line with the findings in [2,3,4] that train-time overparameterization improves the performance.
>
> ---
>
> In conclusion:
> * __Purely using BatchNorm alone does not enhance ViT testing performance.__
> * __Relying solely on a single linear projection for the channel idle process results in lower performance.__
>
> The above discussions and experimental results will be included in the revised version. We hope our explanation can resolve the Reviewer's concerns and __respectfully request consideration for a score increase__.
>
> &nbsp;
>
> [1] Yao, Zhuliang, et al. "Leveraging batch normalization for vision transformers." ICCV, 2021.
>
> [2] Vasu, Pavan Kumar Anasosalu, et al. "FastViT: A fast hybrid vision transformer using structural reparameterization." ICCV, 2023.
>
> [3] Vasu, Pavan Kumar Anasosalu, et al. "Mobileone: An improved one millisecond mobile backbone." CVPR, 2023.
>
> [4] Ding, Xiaohan, et al. "Repvgg: Making vgg-style convnets great again." CVPR, 2021.

---

> > ### Comment · Reviewer_ufAp · 2024-11-23
> >
> > Sorry for the late reply and thank you for your response and additional experiments!
> >
> > It seems that in the new setting presented, the Rep. Linear 3 does not have any normalization. (Please correct me if I am wrong.) However, as I mentioned in my initial review, I expected a baseline similar to the form of Figure (b)—specifically Linear projection 3 with a BatchNorm layer but without BatchNorm inference reparameterization.  Therefore, I would suggest conducting experiments where Linear projection 3 has its own BatchNorm layer during training.
> >
> > Would it be possible to provide results for such a baseline?

---

> > > ### Author Response · Authors · 2024-11-23
> > >
> > > We sincerely thank for your response. We would like to conduct the experiment as suggested and will update the results soon. However, with the BatchNorm not being reparamterized, the shortcut layer cannot be reparameterized into the linear projection either. So in the supplementary experiments, we will keep the shortcut in each FFN layer.

---

> > > > ### Comment · Reviewer_ufAp · 2024-11-25
> > > >
> > > > Could you clarify what you mean by:
> > > >
> > > > "However, with the BatchNorm not being reparameterized, the shortcut layer cannot be reparameterized into the linear projection either. So in the supplementary experiments, we will keep the shortcut in each FFN layer."
> > > >
> > > > Thank you in advance for your explanation.

---

> ### Author Response · Authors · 2024-11-20
> **Kind Request for Rebuttal Discussion or Reconsideration of the Score**
>
> We thank Reviewer ufAp for the valuable feedback and insightful suggestions, which have helped us refine and clarify our work. We have carefully addressed all the raised concerns in our response.
>
> We would greatly appreciate it if Reviewer ufAp could provide further feedback. Your input is invaluable to ensuring the quality and clarity of our work. Or, __if our responses have satisfactorily resolved the concerns, we respectfully request reconsideration of the score based on the clarifications and improvements provided__.

---

> ### Author Response · Authors · 2024-11-23
> **Kind Request for Discussion and Reconsideration of the Score**
>
> We sincerely thank the reviewer for the valuable suggestions and insightful comments. In our response, we have carefully addressed all the raised concerns regarding 1) the effect of BatchNorm, and 2) the comparison with the reparameterized model trained from scratch.
>
> If the reviewer has any further questions, __we are glad to join in the discussion__. Otherwise, if our responses have satisfactorily resolved the concerns, __could the reviewer reconsider the score based on our clarifications and responses?__

---

> ### Author Response · Authors · 2024-11-25
>
> We are glad to clarify this statement.
>
> To begin with, as shown in Figure 1(b) and Equation 2, there is a shortcut for each FFN layer as
> $$
>     Z = \text{FFN}(\text{Norm}(Y)) + Y.
> $$
> However, the vanilla FFN incorporates LayerNorm (i.e., $\text{Norm}(\cdot)$=$\text{LN}(\cdot)$) as its normalization method, which is specific to the input feature and cannot be structurally reparameterized into linear projection weights. Consequently, the shortcut component (i.e., $+ Y$) cannot be reparameterized either. To address this, we replace LayerNorm with BatchNorm (i.e., $\text{Norm}$=$\text{BN}(\cdot)$), enabling the reparameterization of both the normalization and the shortcut.
>
> Next, as suggested by the reviewer, the new baseline model should follow the structure in Figure 1(b), but with the channel idle mechanism replaced by a linear projection. This process can be expressed as:
> \begin{equation}
>     Z = \text{Act}(\text{BN}(Y)W^{\text{1}})W^{\text{2}} + \text{BN}(Y)W^{\text{3}} + Y,
> \end{equation}
> where $W^{\text{3}}\in\mathbb{R}^{C\times C}$. __We are now directly training and testing the new baseline model as formulated above.__
>
> Nonetheless, as the reviewer further required that _specifically Linear projection 3 with a BatchNorm layer but without BatchNorm inference reparameterization_, the shortcut $+Y$ in the above baseline model cannot be reparameterized into the projection weights $W^{\text{3}}$ without BatchNorm reparameterized during inference. So we simply keep the BatchNorm and shortcut during testing, which can lead to a bit slower throughput.
>
> In addition, __if the BatchNorms and shortcut in the above model are expected to be reparameterized during testing, we assure the reviewer that the accuracy will not drop after reparameterization while the inference speed and model size should be analogous to our model.__
>
> The experimental results will be updated soon.

---

> > ### Author Response · Authors · 2024-11-26
> > **Supplementary experiment results**
> >
> > The comparison results are presented in the table below. The _"New baseline"_ refers to the setting where the channel idle mechanism is replaced by a $\mathbb{R}^{C\times C}$ linear projection weight with BatchNorm during training. The _"Ours"_ refers to the RePaFormer models trained with $6C^2$ linear projection weights. All the models are trained using the same training recipes as their corresponding RePaFormer variants.
> >
> > To provide a comprehensive comparison, we report both the pre- and post-reparameterization results during inference. It is important to note that __the post-reparameterization structure of _"New baseline"_ models should be the same as that of our RePaFormers__.
> >
> > In addition, for fairness, we re-test all the models on a single A6000 GPU. As a result, the inference throughputs may differ slightly from those reported in the manuscript.
> >
> > |Backbone|Method|Inference Reparam.|#MParam.|Complexity (GMACs)|Throughput (img/s)|Top-1 accuracy|
> > |:-|:-|:-|:-|:-|:-|:-|
> > |__DeiT-Tiny:__||||||
> > ||New baseline|×|__3.5__|__0.8__|3993.4|__64.3%__|
> > ||New baseline|√|__3.5__|__0.8__|__4328.0__|__64.3%__|
> > ||Ours|√|__3.5__|__0.8__|__4328.0__|64.2%|
> > |__DeiT-Small:__||||||
> > ||New baseline|×|__13.2__|__2.9__|1774.2|75.7%|
> > ||New baseline|√|__13.2__|__2.9__|__1983.5__|75.7%|
> > ||Ours|√|__13.2__|__2.9__|__1983.5__|__77.1%__|
> > |__DeiT-Base:__||||||
> > ||New baseline|×|__51.1__|__10.6__|589.8|80.6%|
> > ||New baseline|√|__51.1__|__10.6__|__654.6__|80.6%|
> > ||Ours|√|__51.1__|__10.6__|__654.6__|__81.4%__|
> > |__ViT-Large:__||||||
> > ||New baseline|×|__178.4__|__34.9__|188.9|80.3%|
> > ||New baseline|√|__178.4__|__34.9__|__199.7__|80.3%|
> > ||Ours|√|__178.4__|__34.9__|__199.7__|__82.0%__|
> > |__Swin-Tiny:__||||||
> > ||New baseline|×|__17.5__|__2.6__|966.3|78.0%|
> > ||New baseline|√|__17.5__|__2.6__|__1021.9__|78.0%|
> > ||Ours|√|__17.5__|__2.6__|__1021.9__|__78.5%__|
> > |__Swin-Small:__||||||
> > ||New baseline|×|__29.9__|__5.1__|595.2|79.1%|
> > ||New baseline|√|__29.9__|__5.1__|__654.6__|79.1%|
> > ||Ours|√|__29.9__|__5.1__|__654.6__|__81.6%__|
> > |__Swin-Base:__||||||
> > ||New baseline|×|__52.8__|__9.0__|417.0|80.3%|
> > ||New baseline|√|__52.8__|__9.0__|__452.8__|80.3%|
> > ||Ours|√|__52.8__|__9.0__|__452.8__|__82.6%__|
> > |__LV-ViT-S:__||||||
> > ||New baseline|×|__19.1__|__4.7__|1031.0|81.3%|
> > ||New baseline|√|__19.1__|__4.7__|__1125.6__|81.3%|
> > ||Ours|√|__19.1__|__4.7__|__1125.6__|__81.6%__|
> > |__LV-ViT-M:__||||||
> > ||New baseline|×|__40.1__|__8.8__|582.8|NaN (collapse)|
> > ||New baseline|√|__40.1__|__8.8__|__643.6__|NaN (collapse)|
> > ||Ours|√|__40.1__|__8.8__|__643.6__|__83.5%__|
> >
> > In conclusion:
> >
> > * __Training with $6C^2$ parameters generally achieves better performance than training with $C^2$ parameters.__
> >
> > * __The benefit of training with $6C^2$ parameters signifies as the model size increases.__
> >
> > * __Inference-time reparameterization of BatchNorm and shortcut improves the throughput.__
> >
> > We hope our experiment results can address the reviewer's concerns and respectfully request a reconsideration of the score.

---

> > > ### Comment · Reviewer_ufAp · 2024-11-26
> > >
> > > Thank you for your additional experiments! While the results are appreciated, the improvements over the new baselines appear to be modest, particularly when factoring in the training costs of the RePa models compared to the new baselines with only $C^2$ fully-connected parameters. Additionally, the hyper-parameters of the new baselines may not be tuned, as shown in NaN for LV-ViT-M cases.
> > >
> > > However, the responses have addressed some of my questions. I have decided to increase my rating from 3 to 5.

---

> > > > ### Author Response · Authors · 2024-11-28
> > > > **Response to Reviewer ufAp's Further Concerns (Part 1)**
> > > >
> > > > We thank Reviewer ufAp for the reply and are pleased to know that most of your concerns have been addressed. We would like to answer your further concerns as below:
> > > >
> > > > Firstly, __we respectfully disagree with the assertion that _"the improvements over the new baselines appear to be modest"___. When using Swin-Small and Swin-Base as backbones, training with our method improves top-1 accuracy by 2.5% and 2.3% compared to their _new baseline_ variants, respectively. In particular, our method achieves 1.7% higher top-1 accuracy on ViT-Large compared to the new baseline with only 29.8% more training time (223.3 vs 172.0 GPU$\cdot$hours). __These improvements in the top-1 accuracy are substantial, especially in the context of large ViT models, which clearly demonstrate the effectiveness of our approach__.
> > > >
> > > > Secondly, we believe that the additional training cost is reasonable given the performance gains. The rough total training times (GPU$\cdot$hours) on our HPC for these models are provided in the table below.
> > > >
> > > > |Backbone|Vanilla|RePaFormer|New Baseline|
> > > > |:-|:-|:-|:-|
> > > > |DeiT-Tiny|60.0|72.3|51.3|
> > > > |DeiT-Small|70.7|90.0|63.3|
> > > > |DeiT-Base|93.3|123.3|86.7|
> > > > |ViT-Large|196.3|223.3|172.0|
> > > > |Swin-Tiny|117.3|139.3|105.3|
> > > > |Swin-Small|180.0|215.7|158.7|
> > > > |Swin-Base|210.7|272.0|184.0|
> > > > |LV-ViT-S|159.3|189.3|144.0|
> > > > |LV-ViT-M|209.7|250.3|-|
> > > >
> > > > On average, our method incurs approximately 30\~50% more training time. For comparison, when pre-training the vanilla ViT on the JFT-300M dataset, ViT-Huge requires 2.5k TPUv3-core-days while ViT-Large takes only 0.68k TPUv3-core-days—267.6% more training time for just a 0.8% top-1 accuracy increase (statistics taken from the original ViT paper [1]). In contrast, __the accuracy improvements achieved by our method with significantly less overhead highlight the practical value and efficiency of our approach__.
> > > >
> > > > Thirdly, the new baseline (i.e., using $C^2$ projection weights in the FFN layer during training) introduces instability to the training process within the LV-ViT training framework that incorporates token labelling. When taking LV-ViT-M as the backbone, the loss explodes after only a few epochs (fewer than 5), even when the learning rate is still small during the warm-up phase or a gradient clipping is applied. While exploring methods to stabilize the training of this new baseline with token labelling schema is an intriguing research topic, it lies beyond the scope of this work. __As a result, we cannot agree that the absence of large-scale hyperparameter tuning for this specific new baseline on LV-ViT-M should diminish the contribution of this work__.
> > > >
> > > > In conclusion, the trade-off between accuracy gain and training costs is justified and worthwhile.

---

> > > > ### Author Response · Authors · 2024-12-01
> > > >
> > > > Dear Reviewer ufAp,
> > > >
> > > > We sincerely thank you for your continued engagement in the discussion. In our previous response, we made every effort to address your further concerns and to clarify the key contributions and significance of our work, including:
> > > >
> > > > * Explaining that the accuracy gain of our method is NOT moderate compared to the baseline, and the trade-off on training time is worthwhile.
> > > >
> > > > * Providing additional experimental results on large ViT models, where our method achieves both improved efficiency and increased accuracy.
> > > >
> > > > * Demonstrating the transformative contribution of our work for large-scale foundation models in vision tasks.
> > > >
> > > > __We greatly appreciate your time in carefully reviewing our further response. If it satisfactorily resolves all your concerns, we would be deeply grateful for your support of our work and reconsideration of the score.__
> > > >
> > > > Best regards,
> > > > Authors

---

> ### Author Response · Authors · 2024-11-28
> **Response to Reviewer ufAp's Further Concerns (Part 2)**
>
> In addition to resolving the above concerns, we are excited to share the significant contribution and practical value of our RePaFormer with Reviewer ufAp.
>
> Through discussions with you and other Reviewers, and especially inspired by Reviewer wqPR, we have gained a chance to present the key advantages and application scenarios of our approach: __RePaFormer can significantly speed up large ViT models while even improving accuracy__. This insight demonstrates the __practical value of RePaFormer in accelerating large-scale models without compromising performance, making it an effective solution for large real-world applications requiring both speed and precision__.
>
> To empirically validate the practicality above, we conducted additional experiments on large vanilla ViTs, following Reviewer wqPR's kind suggestion. Both the vanilla and RePaFormer variants of ViT-Large and ViT-Huge are trained from scratch on the ImageNet-1k dataset using the same training recipes with the idle ratio set to 75% by default. The new results, along with those for MLPMixer-l16 reported in the original manuscript, are shown in the table below:
>
> |Model|#MParam.|Complexity (GMACs)|Throughput (img/s)|Top-1 accuracy|
> |:-|:-|:-|:-|:-|
> |ViT-Large|304.3|59.7|124.2|80.3%|
> |RePaViT-Large|__178.4__ (-41.4%)|__34.9__ (-41.5%)|__207.2__ (+66.8%)|__82.0%__|
> |ViT-Huge|632.2|124.3|61.5|80.3%|
> |RePaViT-Huge|__369.6__ (-41.5%)|__72.6__ (-41.6%)|__103.8__ (+68.7%)|__81.4%__|
> |MLPMixer-l16|208.2|44.6|460.0|72.3%|
> |RePaMLPMixer-l16|__82.2__ (-60.5%)|__20.0__ (-55.2%)|__302.7__ (+89.2%)|__72.6%__|
>
> We are thrilled to emphasize that our method not only drastically reduces model size and latency but also achieves HIGHER top-1 accuracy on large models with more than 200M parameters and computational complexities exceeding 40 GMACs. For instance, RePaViT-Large achieves a 1.7% higher top-1 accuracy (82.0% vs 80.3%) while delivering a 66.8% speed gain (207.2 images/second vs 124.2 images/second) compared to the vanilla ViT-Large. __This demonstrates a transformative contribution, as many practical large-scale foundation models for computer vision tasks rely on vanilla ViT as their backbone, such as CLIP [1], SAM [2] and ViT-22B [3].__
>
> To the best of our knowledge, RePaFormer is the first novel method that achieves significant acceleration (\~68%) while having positive gains in accuracy (1\~2%) instead of accuracy drops, on large and huge ViTs. Considering the unprecedented results RePaFormer is getting, we want to point out that this is a disruptive and timely innovation for the community and a significant addition to the large foundation models acceleration toolkit. Since RePaFormer can be both directly applied to larger ViT architectures and combined with other acceleration techniques such as quantization, we believe RePaFormer will catalyze further research and breakthroughs on ViT's speed and accuracy. __We strongly believe that the weight and impact of this work make it best-suited for the prestigious ICLR, and the community will benefit greatly by seeing it soon from this venue.__
>
> We hope our response and plenty of additional experimental results can address all your concerns and demonstrate the significance of our contributions. __We kindly request your strong support by considering a score increase__.
>
> &nbsp;
>
> [1] Radford, Alec, et al. "Learning transferable visual models from natural language supervision." ICML, 2021.
>
> [2] Kirillov, Alexander, et al. "Segment anything." ICCV, 2023.
>
> [3] Dehghani, Mostafa, et al. "Scaling vision transformers to 22 billion parameters." ICML, 2023.

---

### Official Review · Reviewer_nzL3 · 2024-11-03

**Soundness:** 2
**Presentation:** 3
**Contribution:** 2
**Rating:** 5
**Confidence:** 5

**Summary:**

This paper presents a new method for accelerating FFN layers in MetaFormer-structured architectures. The core idea is to combine structured reparameterization and partial channel skipping. Experiments are done on ImageNet classification, self-supervised learning, and dense prediction tasks. The proposed method can accelerate various ViT models with some accuracy drop.

**Strengths:**

1. The proposed method is interesting and technically sound.
2. The problem studied in this paper is critical, as FFN is a big efficiency bottleneck for ViTs.
3. I appreciate seeing results outside ImageNet classification.

**Weaknesses:**

1. This paper lacks direct comparisons with network pruning.
2. This paper lacks an essential baseline, training Figure 1 (c) from scratch.
3. This paper lacks direct comparisons with previous structured reparameterization methods in previous works (e.g., FastViT's design).
4. According to the results, the proposed method still suffers from accuracy drops.

**Questions:**

It seems the proposed method has to be used with BatchNorm. Is there any workaround to avoid using BatchNorm?

---

> ### Author Response · Authors · 2024-11-18
> **Response to Reviewer nzL3 (Part 1)**
>
> We sincerely thank Reviewer nzL3 for recognizing that ___problem studied in this paper is critical___ and ___the proposed method is interesting and technically sound___. We are honoured to hear that multiple reviewers have acknowledged the importance of the problem and the novelty of our method. Regarding the issues raised by Reviewer nzL3, we address them as follows.
>
> &nbsp;
>
> ---
>
> __1. Response to W1__ ___(lacking comparisons with network pruning)___:
>
> First, we would like to qualitatively compare the differences between our method and pruning methods:
>
> * __Different motivations__: Network pruning focuses on removing redundant parameters, creating a sparse network. In contrast, our method structurally combines parameters using linear algebra operations, resulting in a condensed yet structurally regular network.
>
> * __Different requirements and implementation difficulties__: Network pruning methods produce sparse networks. To fully utilize the sparsity introduced by pruning, specialized hardware (e.g., accelerators that support sparse computations) or software libraries are needed. This adds complexity to deployment and maintenance. On the contrary, the reparameterized RePaFormers are structurally regular and can be efficiently adopted on general-purpose hardware without specialized support.
>
> * __Different generalizabilities__: Different network architectures have varying sensitivity to pruning; some models may not be suitable for pruning or may not benefit significantly from it. In most cases, dedicated training processes or pruning strategies need to be designed for different backbones. However, our RePaFormer method is generally applicable to all MetaFormer-structured models and can be seamlessly integrated into them.
>
> Second, we would like to provide quantitative analysis by comparing our method with state-of-the-art pruning methods for ViTs in terms of both latency drop and accuracy in the table below:
>
> |Method|Latency drop|Top-1 accuracy|
> |:-|:-|:-|
> |__DeiT-Base:__|||
> |PRACTISE [1]|-14.5%|79.3%|
> |DC-DeiT-B [2]|-16.7%|81.3%|
> |RePaFormer (ours)|-40.3%|81.4%|
> |__Swin-Base:__|||
> |PRACTISE [1]|-14.5%|82.8%|
> |DC-Swin-B [2]|-16.7%|83.8%|
> |RePaFormer (ours)|-33.1%|82.6%|
>
> In addition, we further compare our method with some other representative pruning methods for ViTs regarding computational complexity and accuracy in the table below:
>
> |Model|Complexity (GMACs)|Top-1 accuracy|
> |:-|-|-|
> |__DeiT-Base:__|||
> |PatchSlimming [3]|9.8|81.5%|
> |UVC [4]|8.0|80.6%|
> |WDPruning [5]|9.9|80.8%|
> |X-pruner [6]|8.5|81.0%|
> |RePaFormer (ours)|10.6|81.4%|
> |__Swin-Base:__|||
> |PatchSlimming [3]|9.8|81.5%|
> |UVC [4]|8.0|80.6%|
> |DIMAP2 [7]|10.2|83.4%|
> |RePaFormer (ours)|9.0|82.6%|
>
> As demonstrated in the tables above, our method achieves comparable trade-offs between accuracy and complexity/inference time to network pruning methods.
>
> Last, it is worth emphasizing that __our RePaFormer is parallel to network pruning methods__. As pointed out by Reviewer wqPR, _comparing with those methods is __out of the scope__ of this work_. We would not claim our approach to be more advantageous over network pruning and anticipate they can be utilized together.
>
> After all, we thank the reviewer for this suggestion and will include the comparison in our revised version.
>
> &nbsp;
>
> ---
>
> __2. Response to W2__ ___(lacking an essential baseline)___:
>
> __We thank the reviewer for this contributive comment__. It is important to compare the performance of
>
>    1. training vanilla FFN layers and subsequently reparameterizing them into an efficient structure and
>
>    2. directly training reparameterized FFN layers from scratch
>
> Therefore, we train RePa-DeiT, RePa-Swin and RePa-LV-ViT with a single $C^2$ linear projection weight without the channel idle mechanism from scratch. The training settings are kept consistent with their counterparts outlined in the manuscript. The results are shown in the table below, where the NaN values in the table represent the training crashes.
>
> |Backbone|Ours|Reparameterized (Figure 1(c))|
> |:-|-|-|
> |RePa-DeiT-Tiny|__64.3__|59.6|
> |RePa-DeiT-Small|__77.1__|75.0|
> |RePa-DeiT-Base|__81.4__|_NaN_|
> |RePa-Swin-Tiny|__78.5__|77.1|
> |RePa-Swin-Small|__81.6__|79.3|
> |RePa-Swin-Base|__82.6__|79.6|
> |RePa-LV-ViT-S|__81.6__|_NaN_|
> |RePa-LV-ViT-M|__83.6__|_NaN_|
>
> As shown in the table, __directly training the reparameterized model (as illustrated in Figure 1 (c)) consistently yields a worse performance than our approach__, and occasionally results in training crash. This observation aligns with the findings in [8,9] that training an overparameterized network from scratch achieves a more robust training process and better performance than training a network with fewer parameters. Additionally, we would like to clarify that the crashes in the reparameterized models are primarily due to instability caused by the lack of normalization, leading to unregulated value fluctuations during training.
>
> We will include this baseline in our revised version.

---

> ### Author Response · Authors · 2024-11-18
> **Response to Reviewer nzL3 (Part 2)**
>
> __3. Response to W3__ ___(lacking comparisons with previous structured reparameterization methods)___:
>
> The structural reparameterization in our RePaFormers is different from previous reparameterization methods (e.g., RepVGG [10] or FastViT [9]). We would like to compare RePaFormer and FastViT as follows:
>
> * __Different target components__: FastViT mainly incorporates multi-branch convolutions into the hierarchical ViT network and only applies reparameterization to multi-branch convolutional layers. In other words, FastViT focuses on __accelerating the token mixer layers__ during inference. In contrast, our RePaFormer targets __accelerating the FFN layers__ in ViTs. Our method can be adapted to all models with the FFN layer.
>
> * __Different scopes__: The main scope of FastViT is to construct novel efficient architectures, while our method aims to apply structural reparameterization to FFN layers and accelerate existing ViTs/MetaFormers of all sizes.
>
> * __Different reparameterization solutions__: Another key difference is that FastViT reparameterizes __horizontally__ across parallel convolutional kernels, while RePaFormer reparameterizes __vertically__ on consecutive linear projection weights.
>
>     For instance, FastViT reparameterizes two _parallel_ convolutional branches with kernels $W_1^{\text{Conv}}$ and $W_2^{\text{Conv}}$ by summing them:
>     $$
>     W_{\text{Rep}}^{\text{Conv}} = W_1^{\text{Conv}} + W_2^{\text{Conv}}.
>     $$
>
>     On the contrary, as demonstrated in Equation 6, RePaFormer reparameterizes two _consecutive_ projection weights $W_1^{\text{FFN}}$ and $W_2^{\text{FFN}}$ by multiplying them:
>     $$
>     W_{\text{Rep}}^{\text{FFN}} = W_1^{\text{FFN}} \cdot W_2^{\text{FFN}}.
>     $$
>
>     (In the above example, we omit the BatchNorm and suppose $W_1^{\text{Conv}}$ and $W_2^{\text{Conv}}$ have been padded to the same shape.)
>
> Considering these differences, we think __directly comparing a _dedicatedly designed network architecture_ with a _universal network acceleration method_ is unfair__.  Nonetheless, we do compare a state-of-the-art reparameterization method [11] within the same scope in our paper.
>
> We still appreciate the reviewer for this comment and will include the above comparison in our revised version.
>
> &nbsp;
>
> ---
>
> __4. Response to W4__ ___(the proposed method still suffers from accuracy drops)___:
>
> We respectfully disagree with this weakness.
>
> * __Accuracy is indeed preserved post-reparameterization__: As shown in Table 1, the accuracy remains unchanged before and after reparameterization, while the inference speed significantly improves. This demonstrates that the reparameterization process does not lead to accuracy degradation. In addition, we have provided the resource codes in our supplementary materials, which can validate this as well.
>
> * __Accuracy gap explanation__: The accuracy gap between our method and vanilla models arises because vanilla models do not have idling channels in activation. Training models with significantly fewer non-linearities inherently leads to a performance drop. This trade-off is common across various model compression methods, where increased efficiency often comes at the cost of some accuracy loss.
>
> * __Improved performance with reduced idle ratios__: As shown in Table 4, reducing the idle ratio further improves performance. For example, with a 25% idle ratio, RePa-Swin-Base achieves 83.7% top-1 accuracy, surpassing the vanilla Swin-Base's 83.5% accuracy. This highlights the adaptability of our method to different idle ratios while maintaining competitive accuracy.
>
> &nbsp;
>
> ---
>
> We hope our explanations adequately address the weaknesses raised by Reviewer nzL3. We sincerely appreciate the Reviewer's helpful suggestions and recognition of the significance of our work. We respectfully request reconsideration of the score.
>
> &nbsp;
>
> [1] Wang, Guo-Hua, and Jianxin Wu. "Practical network acceleration with tiny sets." CVPR, 2023.
>
> [2] Zhang, Hanxiao, et al. "Dense Vision Transformer Compression with Few Samples." CVPR, 2024.
>
> [3] Tang, Yehui, et al. "Patch slimming for efficient vision transformers." CVPR, 2022.
>
> [4] Yu, Shixing, et al. "Unified visual transformer compression." ICLR, 2022.
>
> [5] Yu, Fang, et al. "Width & depth pruning for vision transformers." AAAI, 2022.
>
> [6] Yu, Lu, and Wei Xiang. "X-pruner: explainable pruning for vision transformers." CVPR, 2023.
>
> [7] He, Yang, et al. "Data-independent Module-aware Pruning for Hierarchical Vision Transformers." ICLR, 2024.
>
> [8] Vasu, Pavan Kumar Anasosalu, et al. "Mobileone: An improved one millisecond mobile backbone." CVPR, 2023.
>
> [9] Vasu, Pavan Kumar Anasosalu, et al. "FastViT: A fast hybrid vision transformer using structural reparameterization." ICCV, 2023.
>
> [10] Ding, Xiaohan, et al. "Repvgg: Making vgg-style convnets great again." CVPR, 2021.
>
> [11] Guo, Jialong, et al. "SLAB: Efficient Transformers with Simplified Linear Attention and Progressive Re-parameterized Batch Normalization." ICML, 2024.

---

> ### Author Response · Authors · 2024-11-20
> **Kind Request for Rebuttal Discussion or Reconsideration of the Score**
>
> We thank Reviewer nzL3 for the valuable feedback and insightful suggestions, which have helped us refine and clarify our work. We have carefully addressed all the raised concerns in our response.
>
> We would greatly appreciate it if Reviewer nzL3 could provide further feedback. Your input is invaluable to ensuring the quality and clarity of our work. Or, __if our responses have satisfactorily resolved the concerns, we respectfully request reconsideration of the score based on the clarifications and improvements provided__.

---

> ### Author Response · Authors · 2024-11-23
> **Kind Request for Discussion and Reconsideration of the Score**
>
> We sincerely thank the reviewer for the valuable suggestions and insightful comments. In our response, we have carefully addressed all the raised concerns regarding 1) the comparison with network pruning, 2) the comparison with the reparameterized model trained from scratch, 3) the comparison with FastViT and 4) the accuracy drop confusion.
>
> If the reviewer has any further questions, __we are glad to join in the discussion__. Otherwise, if our responses have satisfactorily resolved the concerns, __could the reviewer reconsider the score based on our clarifications and responses?__

---

> ### Author Response · Authors · 2024-11-26
> **Another Respectful Request for Discussion and Reconsideration of the Score**
>
> Dear Reviewer nzL3,
>
> We sincerely appreciate your time and effort in reviewing our work, especially pointing out that ___the problem studied in this paper is critical___. In the previous rebuttal, we have made every effort to appropriately address all your concerns.
>
> As for your question on ___is there any workaround to avoid using BatchNorm___, we would like to clarify that __LayerNorm is specific to the input feature and is therefore not structurally reparameterizable.__ To enable the reparameterization of both the normalization layer and shortcut, an input-agnostic normalization method (e.g., BatchNorm) is needed. However, exploring ways to reparameterize LayerNorm can be an interesting direction for future research.
>
> In the end, we kindly request your feedback on our rebuttal. We are willing to answer any further questions during the remaining discussion period.
>
> Best regards,
>
> Authors

---

> > ### Comment · Reviewer_nzL3 · 2024-11-26
> >
> > I appreciate the authors' detailed responses addressing some of my concerns. I have raised my rating from 3 to 5 accordingly.
> >
> > Current results regarding the comparison with network pruning are a bit weak. I do not see clear advantages of the proposed method over network pruning, especially given that the proposed method is much more complicated. Although combining network pruning with the proposed method is possible, I am unsure if it can work without experimental results. Given the current results, I am unclear about the practical value of the proposed method.

---

> > > ### Author Response · Authors · 2024-11-28
> > > **Response to Reviewer nzL3's Further Concerns (Part 2)**
> > >
> > > To the best of our knowledge, RePaFormer is the __first novel method _(orthogonal to network pruning, quantization and distillation)_ that achieves significant acceleration (\~68%) while having positive gains in accuracy (1\~2%) instead of accuracy drops, on large and huge ViTs__. Considering the unprecedented results RePaFormer is getting, we want to point out that this is a disruptive and timely innovation for the community and a significant addition to the large foundation models acceleration toolkit. Since RePaFormer can be both directly applied to larger ViT architectures and combined with other acceleration techniques such as quantization, we believe RePaFormer will catalyze further research and breakthroughs on ViT's speed and accuracy. __We strongly believe that the weight and impact of this work make it best-suited for the prestigious ICLR, and the community will benefit greatly by seeing it soon from this venue.__
> > >
> > > We hope our response can address all your concerns and demonstrate the significance of our contributions. We kindly request your strong support by considering a score increase.
> > >
> > > &nbsp;
> > >
> > > [1] Radford, Alec, et al. "Learning transferable visual models from natural language supervision." ICML, 2021.
> > >
> > > [2] Kirillov, Alexander, et al. "Segment anything." ICCV, 2023.
> > >
> > > [3] Dehghani, Mostafa, et al. "Scaling vision transformers to 22 billion parameters." ICML, 2023.

---

> > > ### Author Response · Authors · 2024-12-01
> > >
> > > Dear Reviewer nzL3,
> > >
> > > We sincerely thank you for your continued engagement in the discussion. In our previous response, we made every effort to address your further concerns and to clarify the key contributions and significance of our work, including:
> > >
> > > * Explaining that our method is simpler to implement than network pruning.
> > >
> > > * Providing additional experimental results on large ViT models, where our method achieves both improved efficiency and increased accuracy.
> > >
> > > * Demonstrating the transformative contribution of our work for large-scale foundation models in vision tasks.
> > >
> > > __We greatly appreciate your time in carefully reviewing our further response. If it satisfactorily resolves all your concerns, we would be deeply grateful for your support of our work and reconsideration of the score.__
> > >
> > > Best regards,
> > > Authors

---

> ### Author Response · Authors · 2024-11-28
> **Response to Reviewer nzL3's Further Concerns (Part 1)**
>
> We sincerely appreciate Reviewer nzL3 for carefully considering our rebuttal responses and for raising the score. We are pleased to hear that all other concerns have been appropriately addressed, except for the practical value of our method. The further suggestion to clearly articulate the advantages of our method over network pruning, as well as its practical value, is greatly appreciated and will further enhance the clarity and impact of our contributions.
>
> Firstly, __we respectfully disagree with the statement in your reply that _"the proposed method is much more complicated (than network pruning)"___. On the contrary, our method is both simple and effective. Unlike network pruning, which requires a detailed analysis of the network structure and parameter redundancy, our approach only involves straightforward modifications: replacing LayerNorm with BatchNorm and keeping certain channels inactivated in the FFN layer. Additionally, our method can be easily integrated into MetaFormer-structured models without significant architectural changes. Therefore, we conclude that __our method is generally simpler to implement and more adaptable than network pruning__.
>
> We would like to show pseudocodes in PyTorch style:
>
> ```python
> class Mlp(nn.Module):
>     def __init__(self, dim_in, dim_hidden, dim_out, act_layer, idle_ratio=0.75):
>         super().__init__()
>         self.norm1 = nn.BatchNorm1d(dim_in)
>         self.fc1 = nn.Linear(dim_in, dim_hidden)
>         self.norm2 = nn.BatchNorm1d(dim_hidden)
>         self.fc2 = nn.Linear(dim_hidden, dim_out)
>         self.act = act_layer()
>         self.idle_channels = int(dim_hidden * idle_ratio)
>
>     def forward(self, x):
>         x = self.norm1(x.transpose(-1,-2)).transpose(-1, -2)
>         x = self.fc1(x)
>
>         # Activation with channel idle mechanism
>         mask = torch.zeros_like(x, dtype=torch.bool)
>         mask[:, :, self.idle_channels:] = True
>         x = torch.where(mask, self.act(x), x)
>
>         x = self.norm2(x.transpose(-1,-2)).transpose(-1, -2)
>         x = self.fc2(x)
>         return x
> ```
>
> Secondly, through discussions with you and other Reviewers, and especially inspired by Reviewer wqPR, we have gained a chance to further present the key advantages and application scenarios of our approach: __RePaFormer can significantly speed up large ViT models while even improving accuracy__. This insight demonstrates the __practical value of RePaFormer in accelerating large-scale models without compromising performance, making it an effective solution for large real-world applications requiring both speed and precision__.
>
> To empirically validate the practicality above, we conducted additional experiments on large vanilla ViTs, following Reviewer wqPR's kind suggestion. Both the vanilla and RePaFormer variants of ViT-Large and ViT-Huge are trained from scratch on the ImageNet-1k dataset using the same training recipes with the idle ratio set to 75% by default. The new results, along with those for MLPMixer-l16 reported in the original manuscript, are shown in the table below:
>
> |Model|#MParam.|Complexity (GMACs)|Throughput (img/s)|Top-1 accuracy|
> |:-|:-|:-|:-|:-|
> |ViT-Large|304.3|59.7|124.2|80.3%|
> |RePaViT-Large|__178.4__ (-41.4%)|__34.9__ (-41.5%)|__207.2__ (+66.8%)|__82.0%__|
> |ViT-Huge|632.2|124.3|61.5|80.3%|
> |RePaViT-Huge|__369.6__ (-41.5%)|__72.6__ (-41.6%)|__103.8__ (+68.7%)|__81.4%__|
> |MLPMixer-l16|208.2|44.6|460.0|72.3%|
> |RePaMLPMixer-l16|__82.2__ (-60.5%)|__20.0__ (-55.2%)|__302.7__ (+89.2%)|__72.6%__|
>
> We are thrilled to emphasize that our method not only drastically reduces model size and latency but also achieves __HIGHER__ top-1 accuracy on large models. For instance, RePaViT-Large achieves a 1.7% higher top-1 accuracy (82.0% vs 80.3%) while delivering a 66.8% speed gain (207.2 images/second vs 124.2 images/second) compared to the vanilla ViT-Large. __This demonstrates a transformative contribution, as many practical large-scale foundation models for computer vision tasks rely on vanilla ViT as their backbone, such as CLIP [1], SAM [2] and ViT-22B [3]__.

---

### Official Review · Reviewer_wqPR · 2024-11-05

**Soundness:** 3
**Presentation:** 3
**Contribution:** 3
**Rating:** 6
**Confidence:** 4

**Summary:**

This paper studies the feed-forward network (FFN) layers in MetaFormer architecture and finds it play a significant role in introducing latencies. Based on this observation, the authors propose ReParameterizable Vision Transformers (RePaFormers) with the structural reparameterization technique and reduces the latency remarkably with minor sacrifice in accuracy. Extensive experiments on various tasks and datasets demonstrate the effectiveness of the proposed method.

**Strengths:**

1. The idea of applying structural reparameterization in FFN layers in great and it brings actually speedup in GPU latency.
2. The experimental results are extensive as the method is validated not only on classification tasks, but on downstream tasks and self-supervised learning setting as well, which highlight the generalization ability of the proposed method.
3. Overall, the paper is clear written and well-organized.

**Weaknesses:**

1. Advantage over other model compression techniques. The goal of the proposed method is to increase the efficiency of current architectures, while it can be also realized by other model compression techniques, including pruning, distillation or quantization. The reviewer understands that comparing with those methods is out of the scope of this work, but it would be great if the authors could provide some justification of the advantage of the proposed method. For example, whether the proposed method is more generalizable across model architectures, easier to implement, or has less impact on accuracy compared to other techniques.
2. Performance gap with self-supervised baselines. It is noticeable that the performance gap with self-supervised baselines is larger than the gap in supervised learning, which may hinder its application on foundation models. Meanwhile, it is also unknown that if the proposed method will brings negative impact on the generalization ability of self-supervised learning methods, and experiments on downstream tasks like fine-grained classification may validate this point.
3. Performance gap at downstream tasks. It is also noteworthy that the performance gap at dense prediction tasks is non-negligible compared to the gap in classification tasks. It would better if the authors could provide some explanations or analysis.

**Questions:**

Apart from the questions in weakness, the reviewer has two additional questions:

1. Training costs. What would be the training time of the proposed method compared to the vanilla version?
2. The authors have mentioned in the abstract that 'improvements in speed and accuracy are expected on even larger models', which may not be convincing enough, as the improvements in accuracy should be supported by empirical results.

---

> ### Author Response · Authors · 2024-11-17
> **Response to Reviewer wqPR (Part 1)**
>
> We sincerely appreciate Reviewer wqPR for all the contributive comments and support to our innovative work, especially highlighting that our ___idea is great and the experiments are extensive___.
>
> &nbsp;
>
> ---
>
> __1. Response to W1__ ___(justification of the advantage of the proposed method)___:
>
> We thank the reviewer for acknowledging that _direct comparisons with other network compression methods are_ ___out of the scope of this work___. And we admit that highlighting the significance of our method and demonstrating the advantages over other compression methods is necessary in our work.
>
> First, we would like to emphasize the significance of our method as follows:
>
> 1. __Pioneering the acceleration of FFN layers__: The FFN layer in MetaFormer-structured models is essential and indispensable. Specifically, [1] empirically demonstrates that the token mixer (e.g., self-attention or convolution) can be replaced by simpler operations (e.g., pooling), while the FFN layer should remain indispensable. Besides, [2] investigates the critical role of the FFN layer in the general Transformer architecture.
>
>    However, despite its importance, the FFN layer constitutes a significant portion of the inference time, as shown in our study, yet there has been little research on accelerating this component. To the best of our knowledge, this work is the first to specifically target the acceleration of the FFN layer.
>
> 2. __Generalizability to various MetaFormer-structured models__: The MetaFormer structure has been widely validated as the de facto architecture for computer vision tasks by various works [3,4]. Our method can be seamlessly integrated into these models without requiring specialized design modifications. This generalizability has been thoroughly demonstrated through the results presented in Tables 1 and 2.
>
> 3. __Scalability to larger models__: As shown in Table 2, our method achieves greater speed gains, smaller model sizes, and narrower performance gaps as the model size increases within the same architecture. when scaling from DeiT-Tiny to DeiT-Base, the accuracy drop decreases significantly from 7.9% to 0.4%, while the inference speed gain increases from 32.6% to 67.5%.
>
> Next, when compared with other compression methods, our method has the following advantages:
>
> 1. __Hardware friendly__: Our reparameterized model is dense and structurally regular, making it efficient to run on general-purpose hardware without requiring specialized hardware support. On the contrary, quantization needs support for low-precision computations, and pruning usually requires support for sparse matrix operations.
>
> 2. __Easy to implement__: Our RePaFormer is compatible with existing training and deployment pipelines and can be seamlessly embedded into existing MetaFormer-structured models. There is no need for specific adjustments in the training framework like quantization, distillation or pruning.
>
> In addition, we would also like to compare our method with a state-of-the-art pruning method, DC-ViT [5] (CVPR24), since pruning methods are closer to our approach. Given that DC-ViT has not released the code, we only compare the latency drop reported in the paper. As the table below shows, our method delivers comparable performance to the state-of-the-art pruning method and achieves a better trade-off on larger backbones.
>
> |Model|Latency drop|Top-1 acc. (%)|
> |:-|-|-|
> |DC-ViT-T|-16.5%|__64.4__|
> |RePa-ViT-T|__-24.6%__|64.3|
> |DC-ViT-S|-16.7%|__78.6__|
> |RePa-ViT-S|__-35.3%__|77.1|
> |DC-DeiT-B|-16.7%|81.3|
> |RePa-DeiT-B|__-40.3%__|__81.4__|
>
> In conclusion, the suggestion to justify our advantages over existing methods is very constructive and insightful. We will take this suggestion and include the discussion in our revised version. Nonetheless, __it is worth noting that our method is parallel to these model compression methods and can be combined with them to achieve further acceleration__. We hope our approach provides a promising direction for accelerating large ViT models.

---

> ### Author Response · Authors · 2024-11-17
> **Response to Reviewer wqPR (Part 2)**
>
> __2. Response to W2__ ___(the performance gap with self-supervised baselines is larger)___:
>
> We thank the Reviewer for this insightful comment. However, applying our method to self-supervised learning models is an extended exploration included in this paper. The primary purpose of this experiment is to validate the key characteristic of our model—__the inference speed gain increases and the accuracy gap narrows as the model size grows__—still holds for self-supervised learning. Thus, we simply train our RePaFormers with the same self-supervised training schema as the baselines without any optimizations. We acknowledge the importance of employing our method in self-supervised learning baselines and plan to explore this direction in-depth in our future work, with a specific focus on generalizability as suggested.
>
> After all, the main focus of our work still remains on supervised learning, where our method achieves significant success. This alone already represents a substantial contribution of our approach.
>
> &nbsp;
>
> ---
>
> __3. Response to W3__ ___(the performance gap at dense prediction tasks is non-negligible)___:
>
> We interpret this comment as asking _why the performance gap on dense prediction tasks is very different from the performance gap on classification task_:
>
>  1. First, we kindly argue that dense prediction tasks and classification tasks use distinct evaluation metrics—mean average precision (mAP) for dense prediction and accuracy for classification. __Directly comparing the different metric gaps is thus inherently unfair__.
>
> 2. Second, even if such a comparison is made, __the performance gaps for dense prediction tasks align closely with those for classification tasks__. Specifically, the accuracy gaps for RePa-Swin-Small and RePa-Swin-Base on classification tasks are 1.4% and 0.9%, respectively. Similarly, the mAP gaps for RePa-Swin-Small and RePa-Swin-Base on object detection tasks using Mask R-CNN are 0.019 and 0.01, respectively. These results indicate that the performance gaps are comparable, and the trend is consistent: larger models exhibit narrower performance gaps.
>
> 3. Additionally, when incorporating RetinaNet as the dense predictor, our method achieves even higher mAP on the object detection task, further validating its effectiveness.
>
> However, if this comment is asking _why the speed improvement reduces on dense prediction tasks compared to that on the classification task_:
>
> 1. First, we note that __FFN layers still occupy a large portion of the total computational complexity__. Using the Swin Transformer as an example, the theoretical computational complexity of a single MHSA layer is $O(4hwC^2+2M^2hwC)$ while the corresponding FFN layer complexity is $O(8hwC^2)$. Thus, only when $M^2>2C$ does the MHSA layer complexity exceed that of the FFN layer.
>
>    However, in the Swin Transformer architecture, the window size $M$ is fixed at 7 while the channel dimension $C$ is at least 96 and increases across layers. Therefore, the FFN layers consistently represent a larger portion of the theoretical computational complexity.
>
> 2. __The key factor is the increased computational demand of tensor operations__. In fact, as the input resolution increases in dense prediction tasks, the processing time for several tensor operations (e.g., reshaping, copying, and concatenating) also rises significantly. For instance, each Swin Transformer layer operates two window-reshaping operations, one window-shifting operation and one mask-copying operation, which become non-negligible when the input resolution reaches 800$\times$800.
>
>     This reduced speed gain on dense prediction tasks is a common challenge, as evidenced in similar work. For example, our main state-of-the-art competitor, SLAB [6], achieves even smaller speed improvements on dense prediction tasks with the same predictor than our RePaFormer.
>
> &nbsp;
>
> ---
>
> __4. Response to Q1__ ___(what would be the training time of the proposed method compared to the vanilla version)___:
>
> We have provided the estimated training cost of each model on the ImageNet-1K dataset using 16 NVIDIA H100 GPUs in terms of the GPU hours in the table below:
>
> |Backbone | Vanilla | RePaFormer|
> |:-|-|-|
> |DeiT-Tiny|60.0|72.3|
> |DeiT-Small |70.7|90.0|
> |DeiT-Base |93.3|123.3|
> |Swin-Tiny |117.3|139.3|
> |Swin-Small |180.0|215.7|
> |Swin-Base |210.7|272.0|
> |LV-ViT-S |159.3|189.3|
> |LV-ViT-M |209.7|250.3|
> |PoolFormer-s12 |74.7|86.7|
> |PoolFormer-s24 |132.0|150.7|
> |PoolFormer-s36 |186.7|216.0|
> |MLPMixer-b16 |112.0|138.0|
> |MLPMixer-l16 |233.3|298.7|
>
> The BatchNorm in RePaFormers introduces an increased synchronization time between devices during training, leading to 16~28% longer training time on the ImageNet-1K dataset. However, the training overhead is still less significant compared to distillation methods, which can increase the training time by more than 40%.

---

> ### Author Response · Authors · 2024-11-17
> **Response to Reviewer wqPR (Part 3)**
>
> __5. Response to Q2__ ___(improvements in accuracy should be supported by empirical results)___:
>
> We thank the reviewer for this question. First, we would like to clarify that a more precise claim should be: "__Improvements with greater speed gains and narrower accuracy gaps are expected on larger models__". This claim aligns with the statements in Lines 106-107, 421-423, 464-466 and Tables 2, 5, 6.
>
> In addition, we would like to provide results of RePaFormer when using ViT-Large and ViT-Huge as the backbone. The vanilla models and their RePaFormer versions are __trained from scratch solely on the ImageNet-1K dataset with the same training schema__ outlined in the manuscript for fairness. We set the drop path rate at 0.3 for both models. The experiments are ongoing and the results will be updated later (_update: ViT-Large and ViT-Huge results have been updated_).
>
> |Model |#MParam. | Complexity (GMACs) | Throughput (img/s) | Top-1 accuracy|
> |:-|:-|:-|:-|:-|
> |ViT-Large|304.3|59.7|124.2|80.3%|
> |RePaViT-Large|178.4 (-41.4%)|34.9 (-41.5%)|207.2 (+66.8%)|82.0%|
> |ViT-Huge|632.2|124.3|61.5|80.3%|
> |RePaViT-Huge|369.6 (-41.5%)|72.6 (-41.6%)|103.8 (+68.7%)|81.4%|
>
> &nbsp;
>
> ---
>
> In the end, we sincerely appreciate Reviewer wqPR for all the insightful suggestions. We will emphasize the advantages of our approach in the revised version and include the above experiments and analysis to provide further clarity. And we're willing to answer any further questions. __Given this is a new and novel direction in the efficient ViT domain, we hope to get Reviewer wqPR's strong support by increasing the score.__
>
> &nbsp;
>
> [1] Yu, Weihao, et al. "Metaformer is actually what you need for vision." CVPR, 2022.
>
> [2] Geva, Mor, et al. "Transformer feed-forward layers are key-value memories." EMNLP, 2021.
>
> [3] Zhang, Jiangning, et al. "Rethinking mobile block for efficient attention-based models." ICCV, 2023.
>
> [4] Wang, Ao, et al. "Repvit: Revisiting mobile cnn from vit perspective." CVPR, 2024.
>
> [5] Zhang, Hanxiao, Yifan Zhou, and Guo-Hua Wang. "Dense Vision Transformer Compression with Few Samples." CVPR, 2024.
>
> [6] Guo, Jialong, et al. "SLAB: Efficient Transformers with Simplified Linear Attention and Progressive Re-parameterized Batch Normalization." ICML, 2024.

---

> ### Author Response · Authors · 2024-11-20
> **Kind Request for Rebuttal Discussion or Reconsideration of the Score**
>
> We thank Reviewer wqPR for the valuable feedback and insightful suggestions, which have helped us refine and clarify our work. We have carefully addressed all the raised concerns in our response.
>
> We would greatly appreciate it if Reviewer wqPR could provide further feedback. Your input is invaluable to ensuring the quality and clarity of our work. Or, __if our responses have satisfactorily resolved the concerns, we respectfully request reconsideration of the score based on the clarifications and improvements provided__.

---

> ### Author Response · Authors · 2024-11-23
> **Kind Request for Discussion and Reconsideration of the Score**
>
> We sincerely thank the reviewer for the valuable suggestions and insightful comments. In our response, we have carefully addressed all the raised concerns regarding 1) the advantages of our method, 2) the performance gaps on SSL tasks, 3) the performance gaps on dense prediction tasks, 4) the training cost and 5) the performance on even larger models.
>
> If the reviewer has any further questions, __we are glad to join in the discussion__. Otherwise, if our responses have satisfactorily resolved the concerns, __could the reviewer reconsider the score based on our clarifications and responses?__

---

> > ### Comment · Reviewer_wqPR · 2024-11-24
> > **reply**
> >
> > Thanks for the rebuttal.
> >
> > The reviewer appreciates the efforts made by the authors and I tend to keep my original rating for now.

---

> > > ### Author Response · Authors · 2024-11-25
> > >
> > > Dear Reviewer,
> > >
> > > We sincerely appreciate your reply. __We would like to confirm whether all your concerns have been appropriately responded to and resolved.__ There are still a few days left for discussion, and we are glad to clarify any further questions to get your strong support.
> > >
> > > In addition, we are actively working on an update to the manuscript and will ensure that key new results, insights, and discussions generated during the rebuttal are incorporated after the end of the rebuttal stage.
> > >
> > > Best regards,
> > > Authors

---

> ### Author Response · Authors · 2024-11-28
> **Appreciation for Contributive Review Comments and Discussion**
>
> Dear Reviewer wqPR,
>
> We sincerely appreciate your insightful comments and suggestions, especially the recommendation to empirically prove the effectiveness of our method on large and huge ViTs. Through the discussion with you and other Reviewers, we have gained a chance to present the key advantages and application scenarios of our approach: __RePaFormer can significantly speed up large ViT models while even improving accuracy__. This insight demonstrates the practical value of RePaFormer __in accelerating large-scale models without compromising performance, making it an effective solution for large real-world applications requiring both speed and precision__.
>
> We would like to share the results on ViT-Large and ViT-Huge, along with those for MLPMixer-l16 reported in the original manuscript, in the table below. Our method not only drastically reduces model size and latency but also achieves HIGHER top-1 accuracy on large models with more than 200M parameters and computational complexities exceeding 40 GMACs.
>
> |Model|#MParam.|Complexity (GMACs)|Throughput (img/s)|Top-1 accuracy|
> |:-|:-|:-|:-|:-|
> |ViT-Large|304.3|59.7|124.2|80.3%|
> |RePaViT-Large|__178.4__ (-41.4%)|__34.9__ (-41.5%)|__207.2__ (+66.8%)|__82.0%__|
> |ViT-Huge|632.2|124.3|61.5|80.3%|
> |RePaViT-Huge|__369.6__ (-41.5%)|__72.6__ (-41.6%)|__103.8__ (+68.7%)|__81.4%__|
> |MLPMixer-l16|208.2|44.6|460.0|72.3%|
> |RePaMLPMixer-l16|__82.2__ (-60.5%)|__20.0__ (-55.2%)|__302.7__ (+89.2%)|__72.6%__|
>
> To the best of our knowledge, RePaFormer is the __first novel method (orthogonal to network pruning, quantization and distillation) that achieves significant acceleration (\~68%) while having positive gains in accuracy (1\~2%) instead of accuracy drops, on large and huge ViTs__. Considering the unprecedented results RePaFormer is getting, we want to point out that this is a disruptive and timely innovation for the community and a significant addition to the large foundation models acceleration toolkit. Since RePaFormer can be both directly applied to larger ViT architectures and combined with other acceleration techniques such as quantization, we believe RePaFormer will catalyze further research and breakthroughs on ViT's speed and accuracy. __We strongly believe that the weight and impact of this work make it best-suited for the prestigious ICLR, and the community will benefit greatly by seeing it soon from this venue.__
>
> We would like to sincerely thank you once again for your thoughtful review and constructive discussion. We hope to receive your strong support through consideration of a score increase.
>
> Best regards,
> Authors

---

### Official Review · Reviewer_NBjZ · 2024-11-05

**Soundness:** 3
**Presentation:** 3
**Contribution:** 2
**Rating:** 5
**Confidence:** 4

**Summary:**

This work proposes a reparameterization technique for vision transformers (ViTs) to improve their test-time efficiency. It achieves this by leaving some channels idle, which are not passed through activation functions and can thus be merged at inference time. Experiments show that the proposed method can notably reduce the latency of ViT-based classification models at the cost of some accuracy loss.

**Strengths:**

1. This paper is clearly written and easy to follow.

2. The proposed method is motivated by a comprehensive latency analysis.

3. Experiments demonstrate that the proposed method significantly improves the throughput of ViT-based classification models.

**Weaknesses:**

1. The main concern with this paper is the significant accuracy drop induced by the reparameterization. As shown in Table 2, the throughput improvement comes at the cost of a substantial accuracy loss, such as -7.9% on DeiT-Tiny and -6.7% on PoolFormer-s12. It appears that the proposed method scales poorly to smaller models, and simpler compression techniques like pruning might be a better option for model acceleration.

2. Another major concern is the lack of analysis and comparison with other reparameterization strategies. Specifically, it is unclear why the proposed method is preferable to RepVGG-style multi-branch reparameterization, as leaving some channels idle without passing through activation functions can be considered a special case of a dual-branch structure. The authors should analyze the underlying reasons and key differences that make the proposed method distinct.

3. The experimental benchmarks are insufficient. A comparison with (1) vanilla reparameterization techniques (e.g., RepVGG-style multi-branch structure) and (2) other compression methods that offer different accuracy-efficiency trade-offs should be included.

4. The latency improvement on dense prediction tasks is small, potentially because FFNs occupy a smaller portion of the runtime for high-resolution inputs.

5. Minor: Tables 1 and 2 have considerable overlap. Retaining only Table 2 should be sufficient.

**Questions:**

My questions and concerns are listed in the weakness section. My main question is why the proposed method is a better choice than RepVGG-style multi-branch reparameterization.

---

> ### Author Response · Authors · 2024-11-16
> **Response to Reviewer NBjZ (Part 1)**
>
> We sincerely thank Reviewer NBjZ for the valuable comments, especially the recognition that _the proposed method is motivated by a comprehensive latency analysis_.
>
> &nbsp;
>
> ---
>
> __1. Response to W2 and Q1__ ___(what makes our method distinct from RepVGG)___
>
> We would like to first respond to the Reviewer's major concern about the differences between our structural reparameterization method and RepVGG-style reparameterization. The differences are threefold:
>
> 1. __Different reparameterization solutions__: The key difference is that RepVGG reparameterizes __horizontally__ across parallel convolutional kernels, while RePaFormer reparameterizes __vertically__ on consecutive linear projection weights.
>
>    For instance, RepVGG reparameterizes two _parallel_ convolutional branches with kernels $W_1^{\text{Conv}}$ and $W_2^{\text{Conv}}$ by summing them:
>    $$
>    W_{\text{Rep}}^{\text{Conv}} = W_1^{\text{Conv}} + W_2^{\text{Conv}}.
>    $$
>
>    On the contrary, as demonstrated in Equation 6, RePaFormer reparameterizes two _consecutive_ projection weights $W_1^{\text{FFN}}$ and $W_2^{\text{FFN}}$ by multiplying them:
>    $$
>    W_{\text{Rep}}^{\text{FFN}} = W_1^{\text{FFN}} \cdot W_2^{\text{FFN}}.
>    $$
>
>    (In the above example, we omit the BatchNorm and suppose $W_1^{\text{Conv}}$ and $W_2^{\text{Conv}}$ have been padded to the same shape.)
>
> 2. __Different target components__: RepVGG and RepVGG-style methods apply reparameterization to multi-branch convolutional layers in CNNs, while our RePaFormer targets FFN layers in ViTs. Their application targets are distinct.
>
> 3. __Different scopes__: Although some previous works [1,2] have attempted to adapt RepVGG-style reparameterization on ViTs by incorporating multi-branch convolutions into the ViT backbone, they only reparameterize the convolutional parts. The main scope of these works is to construct novel mobile-friendly architectures. In contrast, our method is the first to apply structural reparameterization to FFN layers and accelerate existing ViTs/MetaFormers of all sizes.
>
> Moreover, we kindly argue that our __channel idle mechanism cannot be regarded as a special case of a dual-branch structure in RepVGG__. In RepVGG, all branches must be linear so that they can be reparameterized, whereas in our approach, only one branch is linear while the other one is nonlinear.
>
> Nonetheless, __we would not claim our RePaFormer to be more advantageous than RepVGG, as they are parallel approaches solving different problems that can be used simultaneously__. We still appreciate the Reviewer for this comment and will include a detailed comparison with vanilla RepVGG-style reparameterization in our revised version.
>
> &nbsp;
>
> ---
>
> __2. Response to W1__ ___(scaling poorly to smaller models)___:
>
> We thank the Reviewer for this comment, which gives us a chance to emphasize our main claim again. As stated in Lines 28, 106-107, 421-423, 464-466 and Tables 2, the most important characteristic of our method is that __it consistently yields a more substantial speed gain and a much narrower performance gap when the backbone model complexity increases__. We anticipate this method to be increasingly effective on larger Transformer/MetaFormer-based models, aligning well with the growing significance of large foundation models today.
>
> Moreover, in Appendix A.2, we have also observed the mentioned problem and explained that _after applying the channel idle mechanism with a high idle ratio (e.g., 75%), tiny models would lack sufficient non-linear transformations_, which is the major reason for the performance drop on smaller ViTs. It is commonly acknowledged that smaller ViTs are less robust and suffer more severe performance declines when compressed, which holds for both token pruning [3,4] and parameter pruning methods [5,6].
>
> We would like to empirically validate it by presenting the performance of small-size ViT models with various idle ratios:
>
> |Model|Idle ratio ($\theta$)|#MParam.|Complexity (GMACs)|Speed (img/s)|Top-1 acc. (%)|
> |:-|-|-|-|-|-|
> |RePa-DeiT-Tiny||||||
> ||75%|3.5|0.8|4323.8|64.2|
> ||50%|4.4|1.0|3904.2|69.2|
> ||25%|5.3|1.2|3555.1|71.9|
> ||0% (vanilla) |5.7|1.3|3372.2|72.1|
> |RePa-Swin-Tiny||||||
> ||75%|17.5|2.6|1016.3|78.5|
> ||50%|21.8|3.3|927.8|80.5|
> ||25%|26.1|4.0|864.9|81.4|
> ||0% (vanilla)|28.3|4.5|789.8|81.2|
> |RePa-PoolFormer-s12||||||
> ||75%|6.0|0.8|4000.2|70.5|
> ||50%|8.4|1.2|3345.4|74.3|
> ||25%|10.7|1.6|2910.1|76.8|
> ||0% (vanilla)|12.0|1.9|2450.0|77.2|
>
> As the table shows, __our RePaFormers demonstrate narrow performance gaps on smaller models when the idle ratio is less rigorous (i.e., $\theta$ = 25%)__. While scaling to small or tiny-sized models is not the primary focus of this work, our method still shows effectiveness in these cases. In addition, this hyperparameter sensitivity study is insightful and will be added to the revised version.

---

> ### Author Response · Authors · 2024-11-16
> **Response to Reviewer NBjZ (Part 2)**
>
> __3. Response to W3__ ___(not comparing with RepVGG-style methods and other compression methods)___:
>
> 1. __Not comparing with RepVGG-style methods__: As we have explained in the response to W2 and Q1, RepVGG-style methods are quite distinct from our methods and cannot be applied to the FFN layer in ViTs. Directly comparing with methods [1,2] in different scopes is unfair and meaningless. However, we have indeed compared against a state-of-the-art reparameterization method [7] in a similar scope in our manuscript.
>
> 2. __Not comparing with other compression methods__: As pointed out by Reviewer wqPR that _comparing with those (model compression) methods is out of the scope of this work_, our RePaFormer method is parallel to existing model compression methods and can be adapted with them. Thus, we only focus on comparing with the state-of-the-art structural reparameterization method [7] for expediting ViTs.
>
> Despite different scopes, we are willing to provide comparisons between our method and representative pruning methods for ViTs.
>
> 1. We compare with the state-of-the-art pruning method for ViT, DC-ViT [5] (CVPR24). Since DC-ViT has not released the code, we only compare the latency drop reported in the paper:
>
>    |Model|Latency drop|Top-1 acc. (%)|
>    |:-|-|-|
>    |DC-ViT-T|-16.5%|__64.4__|
>    |RePa-ViT-T|__-24.6%__|64.3|
>    |DC-ViT-S|-16.7%|__78.6__|
>    |RePa-ViT-S|__-35.3%__|77.1|
>    |DC-DeiT-B|-16.7%|81.3|
>    |RePa-DeiT-B|__-40.3%__|__81.4__|
>
> 2. We compare our method with other representative pruning methods for ViTs regarding computational complexity and accuracy in the table below:
>
>    |Model|Complexity (GMACs)|Top-1 acc. (%)|
>    |:-|-|-|
>    |__DeiT-Base:__|||
>    |PatchSlimming [8]|9.8|81.5|
>    |UVC [9]|8.0|80.6|
>    |WDPruning [10]|9.9|80.8|
>    |X-pruner [11]|8.5|81.0|
>    |RePaFormer|10.6|81.4|
>    |__Swin-Base:__|||
>    |PatchSlimming [8]|9.8|81.5|
>    |UVC [9]|8.0|80.6|
>    |DIMAP2 [6]|10.2|83.4|
>    |RePaFormer|9.0|82.6|
>
> It is worth noting again that __our RePaFormer is a novel direction of accelerating ViT models, which is parallel to existing compression methods and can be adopted with them simultaneously__.
>
> &nbsp;
>
> ---
>
> __4. Response to W4__ ___(the latency improvement on dense prediction tasks is small)___:
>
> We thank the Reviewer for this insightful comment. We agree that the reduced inference speed gain on dense prediction tasks is partly due to high-resolution inputs; however, we would like to kindly point out that __the key factor is the increased computational demand of tensor operations__.
>
> 1. We note that __FFN layers still occupy a large portion of the total computational complexity__:
>
>    Using Swin Transformer as an example, the theoretical computational complexity of a single MHSA layer is $O(4hwC^2+2M^2hwC)$ while the corresponding FFN layer complexity is $O(8hwC^2)$. Thus, only when $M^2>2C$ does the MHSA layer complexity exceed that of the FFN layer.
>
>    However, in the Swin Transformer architecture, the window size $M$ is fixed at 7 while the channel dimension $C$ is at least 96 and increases across layers. Therefore, the FFN layers consistently represent a larger portion of the computational complexity.
>
> 2. As the input resolution increases, __the processing time for numerous tensor operations (e.g., reshaping, copying, and concatenating) also rises significantly__. Each Swin Transformer layer operates two window-reshaping operations, one window-shifting operation and one mask-copying operation. These on-device tensor operations become non-negligible when the input resolution reaches 800$\times$800.
>
> In fact, this reduced speed gain on dense prediction tasks is a common challenge, as evidenced in similar work. For example, our main state-of-the-art competitor, SLAB [7], achieves even smaller speed improvements on dense prediction tasks with the same predictor than our RePaFormer.
>
> &nbsp;
>
> ---
>
> __5. Response to W5__ ___(overlapping between Tables 1 and 2)___:
>
> Table 1 and Table 2 serve different purposes.
>
> Table 1 illustrates that our RePaFormer method is lossless post-reparameterization, assuring users that our method delivers substantial inference speed gains without accuracy drop after reparameterization in practical scenarios.
>
> Table 2 demonstrates that RePaFormer yields a more significant speed gain and a much narrower performance gap when the backbone model complexity increases.
>
> &nbsp;
>
> ---
>
> We sincerely hope our comprehensive explanations and experimental results can address the Reviewer's doubts and __respectfully request a reconsideration of the score__.

---

> ### Author Response · Authors · 2024-11-16
> **Response to Reviewer NBjZ (Part 3)**
>
> [1] Vasu, Pavan Kumar Anasosalu, et al. "FastViT: A fast hybrid vision transformer using structural reparameterization." ICCV, 2023.
>
> [2] Wang, Ao, et al. "Repvit: Revisiting mobile cnn from vit perspective." CVPR, 2024.
>
> [3] Rao, Yongming, et al. "Dynamicvit: Efficient vision transformers with dynamic token sparsification." NeurIPS, 2021.
>
> [4] Xu, Yifan, et al. "Evo-vit: Slow-fast token evolution for dynamic vision transformer." AAAI, 2022.
>
> [5] Zhang, Hanxiao, Yifan Zhou, and Guo-Hua Wang. "Dense Vision Transformer Compression with Few Samples." CVPR, 2024.
>
> [6] He, Yang, and Joey Tianyi Zhou. "Data-independent Module-aware Pruning for Hierarchical Vision Transformers." ICLR, 2024.
>
> [7] Guo, Jialong, et al. "SLAB: Efficient Transformers with Simplified Linear Attention and Progressive Re-parameterized Batch Normalization." ICML, 2024.
>
> [8] Tang, Yehui, et al. "Patch slimming for efficient vision transformers." CVPR, 2022.
>
> [9] Yu, Shixing, et al. "Unified visual transformer compression." ICLR, 2022.
>
> [10] Yu, Fang, et al. "Width & depth pruning for vision transformers." AAAI, 2022.
>
> [11] Yu, Lu, and Wei Xiang. "X-pruner: explainable pruning for vision transformers." CVPR, 2023.

---

> ### Author Response · Authors · 2024-11-20
> **Kind Request for Rebuttal Discussion or Reconsideration of the Score**
>
> We thank Reviewer NBjZ for the valuable feedback and insightful suggestions, which have helped us refine and clarify our work. We have carefully addressed all the raised concerns in our response.
>
> We would greatly appreciate it if Reviewer NBjZ could provide further feedback. Your input is invaluable to ensuring the quality and clarity of our work. Or, __if our responses have satisfactorily resolved the concerns, we respectfully request reconsideration of the score based on the clarifications and improvements provided__.

---

> ### Author Response · Authors · 2024-11-23
> **Kind Request for Discussion and Reconsideration of the Score**
>
> We sincerely thank the reviewer for the valuable suggestions and insightful comments. In our response, we have carefully addressed all the raised concerns regarding 1) the distinction between our method and RepVGG, 2) performance on smaller models, 3) latency improvement on dense prediction tasks and other minors.
>
> If the reviewer has any further questions, __we are glad to join in the discussion__. Otherwise, if our responses have satisfactorily resolved the concerns, __could the reviewer reconsider the score based on our clarifications and responses?__

---

> > ### Comment · Reviewer_NBjZ · 2024-11-26
> > **Further response**
> >
> > Thank the authors for preparing the rebuttal! Although some of my questions have been addressed, I still have the same concern regarding the insufficient experimental results.
> >
> > Specifically, although the authors want to emphasize the narrower performance gap for larger models, another interpretation of the results is that the methodology may not be applicable to all model scales or model types. For example, according to Table 2, even for the large PoolFormer-s36, the resulting RePa-PoolFormer-s36 from the proposed method performs much worse than the vanilla PoolFormer-s24 in terms of the accuracy-efficiency trade-off. This suggests that the proposed method should not be applied to this model type or scale. Similar cases can also be found in Table 2 of the paper.
> >
> > Additionally, the reason I am curious about comparisons with other compression techniques, although orthogonal, is that it is unclear when the proposed method should be used and under what conditions it will be beneficial. Therefore, it is highly desirable for the authors to clearly state the application scenarios where the proposed method is the preferred choice.

---

> ### Author Response · Authors · 2024-11-28
> **Responses to Reviewer NBjZ's Further Concerns**
>
> We sincerely appreciate Reviewer NBjZ's reply and thoughtful suggestions. The recommendation of clearly stating the application scenario is highly valuable and will greatly enhance the clarity and impact of our work.
>
> In our original manuscript, we have done a series of experiments to explore the adaptability of our method across various architectures, such as PoolFormers. However, through discussions with you and other Reviewers, and especially inspired by Reviewer wqPR, we have gained a chance to present the key advantages and application scenarios of our approach: __RePaFormer can significantly speed up large ViT models while even improving accuracy__. This insight demonstrates the __practical value of RePaFormer in accelerating large-scale models without compromising performance, making it an effective solution for large real-world applications requiring both speed and precision__.
>
> To empirically validate the practicality above, we conducted additional experiments on large vanilla ViTs, following Reviewer wqPR's kind suggestion. Both the vanilla and RePaFormer variants of ViT-Large and ViT-Huge are trained from scratch on the ImageNet-1k dataset using the same training recipes with the idle ratio set to 75% by default. The new results, along with those for MLPMixer-l16 reported in the original manuscript, are shown in the table below:
>
> |Model|#MParam.|Complexity (GMACs)|Throughput (img/s)|Top-1 accuracy|
> |:-|:-|:-|:-|:-|
> |ViT-Large|304.3|59.7|124.2|80.3%|
> |RePaViT-Large|__178.4__ (-41.4%)|__34.9__ (-41.5%)|__207.2__ (+66.8%)|__82.0%__|
> |ViT-Huge|632.2|124.3|61.5|80.3%|
> |RePaViT-Huge|__369.6__ (-41.5%)|__72.6__ (-41.6%)|__103.8__ (+68.7%)|__81.4%__|
> |MLPMixer-l16|208.2|44.6|460.0|72.3%|
> |RePaMLPMixer-l16|__82.2__ (-60.5%)|__20.0__ (-55.2%)|__302.7__ (+89.2%)|__72.6%__|
>
> We are thrilled to emphasize that our method not only drastically reduces model size and latency but also achieves HIGHER top-1 accuracy on large models with more than 200M parameters and computational complexities exceeding 40 GMACs. For instance, RePaViT-Large achieves a 1.7% higher top-1 accuracy (82.0% vs 80.3%) while delivering a 66.8% speed gain (207.2 images/second vs 124.2 images/second) compared to the vanilla ViT-Large. __This demonstrates a transformative contribution, as many practical large-scale foundation models for computer vision tasks rely on vanilla ViT as their backbone, such as CLIP [1], SAM [2] and ViT-22B [3].__
>
> In addition, as asked by Reviewer NBjZ, we further summarize the guidelines for adopting our method as follows:
>
>  * __More applicable to models with complex token mixers__: When the idle ratio remains constant (such as 75% in our experiments), RePaFormer performs better with more complex token mixers, as witnessed in Table 2.
>
>  * __More applicable to larger models__: When the backbone architecture and idle ratio are fixed, our method generally achieves narrower accuracy drops, and even accuracy gains, on larger models.
>
>  * __Smaller idle ratios should be leveraged on smaller models__: In our Response (Part 1), we have provided results for smaller models under different idle ratios to illustrate this point. Smaller models are less robust and need more nonlinearities to improve the feature extraction capability.
>
> To the best of our knowledge, RePaFormer is the __first novel method _(orthogonal to network pruning, quantization and distillation)_ that achieves significant acceleration (\~68%) while having positive gains in accuracy (1\~2%) instead of accuracy drops, on large and huge ViTs__. Considering the unprecedented results RePaFormer is getting, we want to point out that this is a disruptive and timely innovation for the community and a significant addition to the large foundation models acceleration toolkit. Since RePaFormer can be both directly applied to larger ViT architectures and combined with other acceleration techniques such as quantization, we believe RePaFormer will catalyze further research and breakthroughs on ViT's speed and accuracy. __We strongly believe that the weight and impact of this work make it best-suited for the prestigious ICLR, and the community will benefit greatly by seeing it soon from this venue.__
>
> We hope our response can address all your concerns and demonstrate the significance of our contributions. We kindly request your strong support by considering a score increase.
>
> &nbsp;
>
> [1] Radford, Alec, et al. "Learning transferable visual models from natural language supervision." ICML, 2021.
>
> [2] Kirillov, Alexander, et al. "Segment anything." ICCV, 2023.
>
> [3] Dehghani, Mostafa, et al. "Scaling vision transformers to 22 billion parameters." ICML, 2023.

---

> ### Author Response · Authors · 2024-12-01
>
> Dear Reviewer NBjZ,
>
> We sincerely thank you for your continued engagement in the discussion. In our previous response, we made every effort to address your further concerns and to clarify the key contributions and significance of our work, including:
>
> * Providing additional experimental results on large ViT models, where our method achieves both improved efficiency and increased accuracy.
>
> * Demonstrating the transformative contribution of our work for large-scale foundation models in vision tasks.
>
> * Summarizing the guidelines for employing our method on different model architectures and different model sizes.
>
>  __We greatly appreciate your time in carefully reviewing our further response. If it satisfactorily resolves all your concerns, we would be deeply grateful for your support of our work and reconsideration of the score.__
>
> Best regards,
> Authors

---

### Meta-Review · Area_Chair_dTod · 2024-12-08

**Metareview:**

The submission introduces RePaFormer, a structural reparameterization method for FFN layers in Vision Transformers (ViTs), with the promise of improving inference speed while minimizing accuracy loss. Despite extensive rebuttals and additional experiments, the paper fails to convincingly demonstrate a clear and significant advantage over existing methods, including network pruning and other model compression techniques. Concerns about generalizability, applicability, and practicality remain unresolved.

Strengths:
* Broad Experimental Scope: The paper includes evaluations across various tasks and model sizes, showing promising speed-ups for large ViTs.
* Improved Efficiency: The method shows notable inference speed gains for large models, sometimes coupled with minor accuracy improvements.

Weaknesses:
* Modest Improvements: The observed improvements are not consistently significant, particularly when accounting for increased training costs and the complexity of implementation.
* Weak Comparison Baselines: Direct comparisons with existing structured reparameterization methods and network pruning are either inadequate or reveal comparable performance, diminishing the claimed advantage of the proposed method.
* Limited Generalizability: The method is less effective on smaller models and dense prediction tasks, restricting its practical applicability.
* Overemphasis on Large Models: The primary benefits are confined to large ViT models, with limited utility for smaller architectures or diverse tasks.
* Training Instability: Issues such as collapsed training for certain backbones highlight potential scalability challenges.

While RePaFormer offers some promising results, its contributions are overshadowed by the lack of clear advantages over existing methods and concerns about practicality and applicability. The reviewers' consensus highlights the need for more robust comparative baselines, refined theoretical framing, and a clearer articulation of its unique value proposition. Based on the current submission and reviews, RePaFormer does not meet the acceptance threshold for ICLR 2025

**Additional Comments On Reviewer Discussion:**

Reviewer Points and Author Responses:

Baseline Comparisons (NBjZ, nzL3, ufAp)
* Concern: Limited comparisons with network pruning, structured reparameterization, and simplified baselines.
* Response: Authors added comparisons with pruning methods and new baselines using simplified architectures. Results showed marginal improvements, primarily for large models. Comparisons were deemed insufficient for demonstrating significant advantages.

Applicability to Smaller Models (NBjZ)
* Concern: Accuracy drops on smaller models and unclear application scenarios.
* Response: Authors provided additional results with smaller idle ratios for small models, showing slight improvements but still lagging behind simpler methods like pruning.

Training Overheads and Stability (ufAp, nzL3)
* Concern: High training costs and instability in some baselines (e.g., LV-ViT-M).
* Response: Authors justified the training costs as reasonable for large models but acknowledged training instability for certain configurations. The reviewers found this explanation insufficient for broader applicability.

Dense Prediction Tasks (NBjZ, wqPR)
* Concern: Reduced speed improvements on dense prediction tasks.
* Response: Authors attributed this to increased tensor operations with high-resolution inputs. While plausible, the explanation did not alleviate concerns about limited practicality for such tasks.

Use of BatchNorm and Reparameterization (ufAp)
* Concern: Unclear advantage of BatchNorm and reparameterization over simpler alternatives.
* Response: Authors presented new baselines with BatchNorm and reparameterization but showed only marginal benefits, raising doubts about the necessity of the proposed complexity.

Novelty and Contribution (all reviewers)
* Concern: Limited practical value, especially given comparable or superior alternatives like pruning.
* Response: Authors emphasized advantages for large models and scalability, but reviewers found the claims incremental.

---

### Decision · Program_Chairs · 2025-01-22

Reject